# Reviving DSP for Advanced Theorem Proving in the Era of Reasoning Models

**Chenrui Cao[1,2,†], Liangcheng Song[1,2,†], Zenan Li[3],**
**Xinyi Le[4], Xian Zhang[1‡], Hui Xue[1], Fan Yang[1]**
[1]Microsoft Research, [2]University of Science and Technology of China,
[3]Department of Computer Science, ETH Zurich, Switzerland, [4]Rice University
{caochenrui, slc1}@mail.ustc.edu.cn, zenan.li@inf.ethz.ch,
rl105@rice.edu, {zhxian, xuehui, fanyang}@microsoft.com

## Abstract

Recent advancements, such as DeepSeek-Prover-V2-671B and Kimina-Prover-Preview-72B, demonstrate a prevailing trend in leveraging reinforcement learning (RL)-based large-scale training for automated theorem proving. Surprisingly, we discover that even without any training, careful neuro-symbolic coordination of existing off-the-shelf reasoning models and tactic step provers can achieve comparable performance. This paper introduces **DSP+**, an improved version of the Draft, Sketch, and Prove framework, featuring a *fine-grained and integrated* neuro-symbolic enhancement for each phase: (1) In the draft phase, we prompt reasoning models to generate concise natural-language subgoals to benefit the sketch phase, removing thinking tokens and references to human-written proofs; (2) In the sketch phase, subgoals are autoformalized with hypotheses to benefit the proving phase, and sketch lines containing syntactic errors are masked according to predefined rules; (3) In the proving phase, we tightly integrate symbolic search methods like Aesop with step provers to establish proofs for the sketch subgoals. Experimental results show that, without any additional model training or fine-tuning, DSP+ solves 80.7%, 32.8%, and 24 out of 644 problems from miniF2F, ProofNet, and PutnamBench, respectively, while requiring lower budget compared to state-of-the-art methods. DSP+ proves `imo_2019_p1`, an IMO problem in miniF2F that is not solved by any prior work. Additionally, DSP+ generates proof patterns comprehensible by human experts, facilitating the identification of formalization errors; For example, eight wrongly formalized statements in miniF2F are discovered. Our results highlight the potential of classical reasoning patterns besides the RL-based training. Code and results are here: `https://github.com/microsoft/DSP-Plus`.

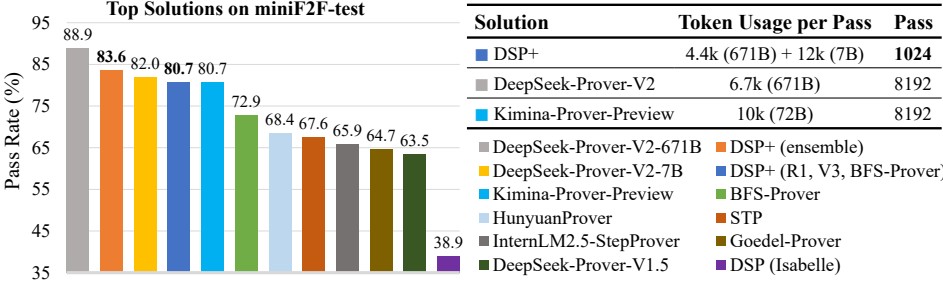

Figure 1: The best achieved results and the inference tokens used of top solutions on miniF2F-test. DSP+, with inference only, can achieve comparable accuracy using fewer tokens.

---

[†]Core and equal contributions. Work done during internship at Microsoft Research.
[‡]Corresponding author.

39th Conference on Neural Information Processing Systems (NeurIPS 2025).

# 1 Introduction

Recently, a trend has emerged to leverage reinforcement learning (RL)-based *large-scale training* to improve the theorem proving capability of large language models (LLMs). Projects such as AlphaProof [1], DeepSeek-Prover-V2 [2], Kimina-Prover-Preview [3], BFS-Prover [4], Goedel-Prover [5], STP-Prover [6], and many others [7, 8, 9, 10, 11] are all dedicated to collecting or synthesizing large-scale formal statements and proofs, which are then used for RL-based LLM training. Coupled with significant human effort and computational resources, remarkable progress has been achieved in challenging benchmarks like miniF2F [12].

Unlike the current trend, the prior *Draft, Sketch and Prove* (DSP) [13] relies on an inference-only framework with three phases to imitate the classical human reasoning patterns of informal-to-formal [14], achieving state-of-the-art at its publication time. Despite the insight, DSP and its subsequent development [15, 5] were limited by the insufficiency of the frontier LLMs and the symbolic search, thus being significantly outperformed by current top solutions, as shown in Figure 1.

In this work, we surprisingly find that by carefully coordinating existing reasoning models and tactic step provers, the DSP framework can be revived as DSP+ to achieve notably higher proving accuracy that is even comparable to current state-of-the-art models tailored for theorem proving. DSP+ achieves this through *fine-grained and integrated* neuro-symbolic enhancements based on three intuitions, which differ from the coarse-grained, phase-independent, and neural-symbolic workflow in the original DSP: (1) Reasoning models generally have a stronger math reasoning capability than non-reasoning models, which can be prompted to generate concise drafts to benefit the sketch phase, with thinking tokens filtered; (2) LLMs can specify hypotheses in the sketch to benefit the proving phase, and most sketch lines after autoformalization are syntactically correct while incorrect ones can be "masked" by simple rules (e.g., replaced with `sorry` or removed directly) to sidestep obstacles in the workflow; (3) Sketch subgoals can be completed using both symbolic search and a weaker step prover like BFS-Prover [4] with an efficient built-in integration.

We instantiate DSP+ in Lean 4 (version v4.17.0-rc1) [16]. In the default setting, DSP+ uses three open-sourced models, QwQ-32B [17], DeepSeek-V3-0324 [18], and BFS-Prover [4] for draft (without human informal proof), sketch, and proving phases, respectively. And DSP+ can achieve comparable accuracy on benchmarks of miniF2F-test (79.5%), ProofNet-test (32.8%), and PutnamBench (24/644), under the search budget of 1024, 128, 128 workflow attempts, respectively. The miniF2F accuracy increases to 80.7% when replacing QwQ-32B with DeepSeek-R1-671B [19], even with fewer total inference tokens spent than those of Kimina-Prover-Preview-72B, as shown in Figure 1.

We also have ablation studies to show that with an ensemble setting of DSP+, namely with different combinations of reasoning models, the accumulative accuracy can be further boosted as 83.6%, 33.9%, 25/644 for miniF2F, ProofNet, and PutnamBench, respectively, and from 40% to 45% for miniF2F/IMO. As shown in Figure 1 and Table 1, the accuracy of DSP+ is either on par with or outperforms prior arts [2, 3, 4, 5, 6, 7, 8, 10], especially given the same search budget. Notably, DSP+ proves `imo_2019_p1`, an IMO problem in miniF2F which is not solved by any prior work (illustrative DSP+ workflow in Appendix H). Moreover, we independently find eight problem statements of miniF2F are wrongly formalized by examining the inconsistent behaviors of DSP+ in subgoal proving (details in Appendix F). We ascribe the power of DSP+ to the synergy of reasoning models, step provers, symbolic search, and the careful neuro-symbolic coordination. Our findings highlight the overlooked potential of existing reasoning models and suggest an efficient, complementary approach to the prevailing trend of RL-based training in theorem proving.

In summary, the contributions of the paper are: (1) We revisit the DSP framework, identifying its underestimated potential and incorporating fine-grained and integrated neuro-symbolic enhancements into its three phases; (2) We develop a system that facilitates flexible and efficient model coordination and ensemble settings, boosting the performance of theorem proving; (3) We conduct comprehensive experiments across various benchmarks to demonstrate the efficacy, efficiency, and synergy of DSP+.

## 2 Background and Related Work

**Reasoning Models.** Reasoning models, like OpenAI o-series, DeepSeek-R1 [19], QwQ-32B [17], are emerging LLMs to demonstrate strong reasoning capability with thinking tokens after a training process of reinforcement learning. For competition-level mathematics such as AIME, which test for

math word problems, reasoning models are achieving scores comparable to top human competitors. However, proving problems that require rigor are still challenging for reasoning models [20, 21].

**Theorem Proving with LLMs.** LLMs' limitation on proving problems necessitates another strand of recent work, which focuses on the proving capability of LLMs with the automatic verification of theorem provers such as Lean [16], Isabelle [22], and Coq [23]. In these approaches, LLMs are required to generate a formal proof, which consists of *tactics* resembling human-written proof steps. The search space for a single tactic is infinite and the tactic is required to link to theorems or axioms in the underlying library, both amplifying the challenges for theorem proving.

Much effort from the community focuses on the foundational setup for theorem proving [24, 25], such as statement and proof synthesis [26, 27, 28, 29, 30, 31, 32], efficient framework and algorithms [33, 34, 35, 36, 37, 38, 37, 15, 39], proof reuse and repair [40, 41]. and library search [42, 43].

Based on these foundational work, recent projects such as AlphaProof [1], DeepSeek-Prover-V2 [2], and Kimina-Prover-Preview [3], and many others [4, 5, 6, 8, 9, 10, 11, 44] build or synthesize large-scale formal corpora for expert iteration, train with large-scale methods such as Rejection Sampling Fine-tuning (RFT) [45] and RL, and scale inference using techniques like Best First Search (BFS) and MCTS, leading to strong results on miniF2F [12], ProofNet [46], and PutnamBench [47].

**Symbolic Search.** Symbolic search is an automatic mechanism widely used in theorem proving, which relies on pattern matching of pre-defined rules. It aims to better leverage the theorem libraries or tactics built upon axioms for vertical or even general domains. Symbolic search is especially useful to close the trivially correct statements or subgoals (e.g., lemmas with `have`) without the tedious effort for the axiom-level rigor. For example, tactics in Lean like `linarith` can generally prove statements related to linear equations and `norm_num` for numeric calculations, while `simp` and `rfl` can work for definitional conversions within a certain computation budget.

And one of the most universal tactics for symbolic search in Lean is `aesop` [43]. Aesop serves as a white-box and highly customizable proof search engine, which allows fine-grained control over the search space and the prioritization of tactics. In Aesop, each step of the search takes a proof state as input and outputs a tactic, through a general tree-based search with the pre-defined prioritization of theorem or tactic candidates. The search process can be configured with a maximum computation budget for a valid proof, which is similar to the timeout in `sledgehammer` [48] of Isabelle.

**Draft, Sketch and Prove (DSP).** To address the challenges of theorem proving, the DSP framework [13] is introduced by synergizing neural and symbolic patterns: (1) to leverage LLMs (i.e., the neural) trained with extensive mathematics corpus for the "intuition" of proof draft; (2) to leverage the symbolic search for bridging the rigor gap between the intuition and the formal proof. Correspondingly, three phases are proposed in *a course-grained and non-integrated setting*, with first two as neural and the last as symbolic while no cross-phase optimization: The draft phase is to generate a natural language proof draft either by referring to human proof or by directly prompting LLMs, which initiates a constrained search. The sketch phase is to generate a hierarchy of subgoals in formal language (i.e., sketch) with omitted proving details, which is to leverage LLMs' capability of autoformalization [27] that interprets the draft from informal to formal. The proving phase is to fulfill the proving details omitted in the sketch by querying the symbolic search to automatically assemble the underlying tactics and theorems. DSP has inspired a sequence of work [49, 50, 40] with its intuitive neural-symbolic synergy. However, as shown in Figure 1, DSP has been surpassed by a wide margin by all the current models, which represent the new era of RL-based large-scale training.

Interestingly, the performance of DSP can be degraded with different settings. As discussed in [15, 5], their re-implementations of DSP in Lean 4 can respectively prove at most 28% and 31% problems of miniF2F with LLMs like OpenAI-o1 and DeepSeek Prover v1.5 [10], showing an accuracy drop w.r.t. the DSP in Isabelle, due to the limited search capability of aesop compared to that of sledgehammer.

## 3 DSP+

In this section, we present DSP+, a DSP framework with *fine-grained and integrated* neuro-symbolic enhancements, where each phase is optimized by both neural models and symbolic rules in consideration of itself and other phases. This is different from the original DSP with draft and sketch phases as neural and the proving phase as symbolic, and with no cross-phase optimizations. For convenience, we use the draft model, the sketch model, and the proving model to represent the LLMs used in the

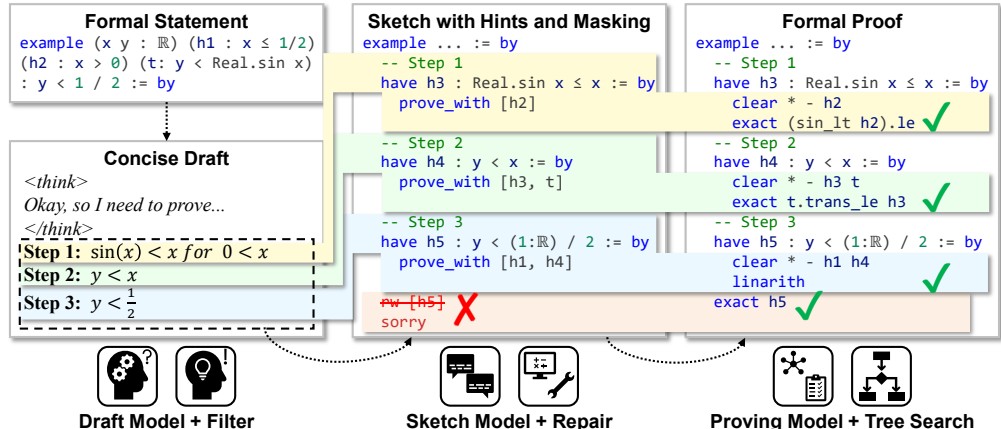

Figure 2: The workflow of DSP+ consists of three phases. Every phase is enhanced with neural models and symbolic rules in consideration of itself and other phases.

corresponding phases. Each of these can be instantiated with any suitable LLM. We provide one-shot prompts for draft models and sketch models, as listed in Appendix G. The workflow, as illustrated in Figure 2, will be explained in the following paragraphs.

## 3.1 Draft Phase with Thinking and Conciseness

As input to DSP+, we directly provide the formal statement to the draft model, instantiated in our work as a reasoning model (e.g., QwQ-32B, DeepSeek-R1). Reasoning models are widely considered more powerful in math reasoning as they can perform deep reasoning, self-reflection, and advanced pattern recognition within the `<think>` tokens, enabling more accurate outputs after *thinking*.

Besides filtering out the thinking tokens, we also prompt the draft model to produce *concise outputs*, focusing only on the key formulas of proof steps. This prevents overloading the sketch phase, which may otherwise suffer from the "lost in the middle" [51] effect. Furthermore, we find it even more important to balance the format and flexibility of model output, as discussed in Section 6.2.

## 3.2 Sketch Phase with LLM Hints and Error Line Masking

We instruct the sketch model to autoformalize [27] the concise informal proof provided by the draft model into a hierarchy of subgoals in formal language, but without the proof for any subgoal. The sketch model is required to explicitly specify the potential supporting hypotheses for each subgoal using a format like (`prove_with [h2]`), as shown in Figure 2. This format acts as structured *hints* to the proving model. We define `prove_with` as a pseudonym for Lean's `sorry`, which denotes that a proposition has not yet been proved but is temporarily assumed to be true; in our implementation, it is augmented with explicit hypothesis usage. The explicit hypothesis specification benefits the proving phase, as we observe that both proving models and symbolic search degrade when faced with overly cluttered subgoals. This is different from DeepSeek-Prover-V2 [2], which leverages either all or none hypotheses in subgoal proving.

The sketch model may still fail to produce a syntactically valid sketch. To address this issue, we introduce an automatic iterative repair process, named *error line masking*. This process first parses the generated Lean code into a hierarchical tree structure based on indentation, after which the Lean compiler identifies all erroneous lines. For each detected error, we prune the corresponding node and its entire subtree from the code tree, optionally appending `sorry` at the original position. We aim to sidestep the re-execution of the sketch phase and retain sketch lines as much as possible, which is different from the proof truncation in DeepSeek-Prover-V1.5 [10].

## 3.3 Proving Phase with Step Prover and Tree Search

For the proving phase to fulfill subgoal proof, we further enhance the symbolic engine with a proving model for a more heuristic search, as envisioned in [13]. Especially, we leverage a *step prover* (e.g., BFS-Prover [4], InternLM2.5-StepProver [52]), which predicts the next tactic given a proof state. We

tightly integrate the step prover with search-based approaches like *tree search* (e.g., Aesop), in which each node in the tactic tree is generated either by the default symbolic engine or by the step prover. This built-in integration can facilitate the independent search for the proof tactics to eliminate every `prove_with` and `sorry` in the sketch given the corresponding proof state (details in Section 4).

## 4 Implementation

In this section, we briefly introduce how we instantiate DSP+, with further details in Appendix A. We choose Lean 4 (v4.17.0-rc1) as the formal language and Aesop [43] for the tree search. We use QwQ-32B [17], DeepSeek-V3-0324 [18], and BFS-Prover [4] as our **default setting** for the draft, the sketch, and the proving models, respectively. To enable the interaction between BFS-Prover and Aesop, we incorporate the Lean Copilot [34] framework, which enables a *built-in* integration of LLM with Aesop's internal logic and circumvents the tedious processes of converting proof states to theorems or external parsing. We also configure Aesop's search space to include tactics proposed by BFS-Prover as well as a few common tactics (e.g., `linarith`), similar to the configuration in [13].

We set the configuration of QwQ-V3-BFS as default according to our observations in early toy experiments, regarding the capability of math reasoning, instruction following, and subgoal proving. In fact, through our later ablation studies (Section 6.3), the default setting is not the optimal given the output randomness and the vast design space. Therefore, for one problem, the workflow of DSP+ can be executed for $k$ times to fully explore the possibility of LLM generation for proving, which contributes to the pass@$k$ accuracy in this work. In addition, besides the default setting, we introduce the **ensemble setting** where different model combinations are used one by one for solving one problem until proven or timeout. For the setting of DSP+ ensemble, DSP+ with different configurations of reasoning models and different pass@$k$ (e.g., pass@1024, pass@128, pass@32) are attached after the default setting for solving one problem. We find that with a moderate search budget, the ensemble setting can further increase the accuracy for solving challenging problems (e.g., IMO, Putnam) given the diversity from different combinations.

We also optimize DSP+ regarding the *modularity* and the *efficiency*, which can benefit both the default and the ensemble settings. For modularity, DSP+ guarantees easy replacement of reasoning models where users only need to update a prompt template according to the LLM interface and serve the model via vLLM [53]. Besides modularity, DSP+ also guarantees efficiency with a pipelined and buffered process for theorem proving: each phase loads the results of the previous phase in the buffer and sends processed results to the buffer for the next phase. And the buffered results are shared for different model combinations to circumvent re-processing.

## 5 Evaluation

### 5.1 Experimental Setup

**QwQ-32B, DeepSeek-V3-0324, and other models.** We use the APIs provided by Microsoft Azure AI Foundry. For QwQ-32B, parameters are set as `temperature` = 0.6, `top-p` = 0.95, and `max_tokens` = 32,768. For DeepSeek-V3-0324, the `temperature` is set to 0.7 with other parameters left as default. We use similar settings for other models.

**BFS-Prover-7B.** We deploy 8×40GB A100 GPUs with one model per GPU using vLLM. The sampling parameters are `temperature` = 1.1, `max_tokens` = 64, `top-p` = 1, n = 8.

**Tree Search.** Tree search is performed on a 96-core CPU hosted on Microsoft Azure, with constraints including a beam width of 4 (selected from 8 sampled tactics) , a tree size limit of 64, and up to 8 search attempts for each subgoal. For details on the sample budget during tree search, please refer to Appendix B. Each proof search and verification process is limited to 2400 seconds. The search process terminates if no available targets, search budget runs out, or the time limit is exceeded. All proofs are verified using Lean 4 (v4.17.0-rc1) with the corresponding Mathlib4 [54].

### 5.2 Benchmarks

**miniF2F [12].** The miniF2F benchmark assesses formal reasoning capabilities in high school mathematics, featuring problems from competitions including AMC, AIME, and IMO. This benchmark

contains balanced splits of 244 validation and 244 test problems, with a curricular focus on algebraic reasoning and number theory. We use the Lean 4 version of the dataset for evaluation.

**ProofNet [46].** Designed for undergraduate-level theorem proving, ProofNet aggregates 371 problems (185 validation, 186 test) spanning core mathematical disciplines: real/complex analysis, linear/abstract algebra, and topology. Similar to miniF2F, we use Lean 4 version for evaluation.

**PutnamBench [47].** PutnamBench provides 1,709 theorem-proving challenges of Putnam Mathematical Competition problems (1962–2023). This benchmark features cross-lingual formalizations aggregated across multiple proof assistants, with our evaluation focusing on the Lean 4 subset, which consists of 644 problems at the time of our evaluation and is extended to 658 problems later.

## 5.3 Results

**Main Results.** Table 1 summarizes the performance of top solutions on the miniF2F-test, ProofNet, and PutnamBench. Due to page limit, we move details into Appendix C. As shown in the table, all top solutions, except DSP+, are models with RL-based large-scale training and categorized as either whole-proof generation or tree search, which complete the proof with a single-pass and with interactions between theorem provers, respectively. By contrast, DSP+ is a hybrid of whole-proof (draft and sketch phases) and tree search (proving phase). With 1024 workflow attempts, DSP+ achieves performance comparable to Kimina-Prover-Preview-72B, which is the frontier model trained extensively via RFT and RL, and evaluated at pass@8192 for miniF2F-test. DSP+ also spends fewer inference tokens compared to Kimina-Prover-Preview-72B, as discussed in Section 6.4.

**Results of DSP+ ensemble**. Furthermore, the accumulative performance of DSP+ ensemble approaches the state-of-the-art accuracy[4] achieved by DeepSeek-Prover-V2-671B for the three benchmarks under the same sample budget, outperforming all other solutions. As detailed in Table 2, the combination of configurations can prove miniF2F-test problems not found by the default configuration, which contributes to the accumulative accuracy. For PutnamBench and ProofNet, our ensemble setting only uses 32 additional workflow attempts, with DeepSeek-R1 as the draft model and other phases intact, resulting in an improved performance on both benchmarks.

**Results of miniF2F/IMO.** As shown in Table 1, we also compare DSP+ and other solutions with available results on the IMO subset of the miniF2F benchmark. Our solution, DSP+ and DSP+ ensemble, can respectively prove 40% and 45% IMO problems with a moderate search budget, on par with Kimina-Prover-Preview-72B and DeepSeek-Prover-V2-671B of pass@8192. Our solution also finds a proof for `imo_2019_p1`, an IMO problem not solved previously, as detailed in Appendix H.

**Case Studies.** We have included the success cases in Appendix J, which show how DSP+ uses Jensen's inequality for a high school competition problem and solves the real analysis problem of the Putnam exam, respectively. We also find the proofs of DSP+ different from those of DeepSeek-Prover-V2, Kimina-Prover-Preview, and BFS-Prover, with details in Appendix M.

## 6 Further Analysis

In this section, we will answer four research questions (RQs) with ablation and case studies:

**RQ1 Synergy.** Does DSP+ synergize reasoning models, step provers, and symbolic search?

**RQ2 Effectiveness.** Does DSP+ benefit from our neural-symbolic enhancements?

**RQ3 Robustness.** Does DSP+ apply to various settings of LLMs and configurations?

**RQ4 Efficiency.** Does DSP+ spend more inference tokens than those with large-scale training?

For the rest of the section, unless specified, we use the default setting as the baseline and set the maximum workflow attempts as 128. And we focus on miniF2F-test for the ablation experiments. In every curve graph, we plot the number of solved problems at different workflow attempts for a stable comparison and include the pass@128 accuracies in the legends for clarity.

---

[4]Issues of Lean 4 v4.9.0-rc1 may negatively impact the accuracy of DeepSeek-Prover-V2 [56].

Table 1: Performance of top solutions across benchmarks (best results among the top 5 in bold). DSP+ achieves performance comparable to frontier models (e.g., Kimina-prover, DeepSeekProver-V2).

| Type | Solution (Model Size) | Sample Budget | miniF2F-test | ProofNet | PutnamBench | miniF2F/IMO |
|---|---|---|---|---|---|---|
| Whole-proof Generation | Goedel-Prover-7B [5] | 32 | 57.6% | 15.2% | 6/644 | – |
| | | 512 | 62.7% | – | 7/644 | – |
| | | 25600 | 64.7% | – | – | – |
| | STP-7B[6] | 3200 | 65.0% | 23.9% | 8/644 | – |
| | | 25600 | 67.6% | 26.9% | – | – |
| | Kimina-Prover-Preview-7B [3] | 1 | 52.5% | – | – | – |
| | | 32 | 63.1% | – | – | – |
| | | 192 | – | – | 10/644[55] | – |
| | | 1024 | 70.8% | – | – | – |
| | Kimina-Prover-Preview-72B [3] | 1 | 52.9% | – | – | – |
| | | 8 | 65.2% | – | – | – |
| | | 32 | 68.9% | – | – | – |
| | | 1024 | 77.9% | – | – | – |
| | | 8192 | **80.7%** | – | – | **40%** |
| | DeepSeek-Prover-V2-7B [2] | 1 | 58.6% | – | – | – |
| | | 32 | 75.6% | 23.0% | 11/658 | – |
| | | 128 | – | 25.4% | 15/658 | – |
| | | 1024 | 79.9% | **29.6%** | **23/658** | – |
| | | 8192 | **82.0%** | – | – | – |
| | DeepSeek-Prover-V2-671B [2] | 1 | 61.9% | – | – | – |
| | | 32 | 82.4% | 30.5% | 22/658 | – |
| | | 128 | – | 33.6% | 33/658 | – |
| | | 1024 | 86.6% | **37.1%** | **49/658** | – |
| | | 8192 | **88.9%** | – | – | **50%** |
| Tree Search | InternLM2.5-StepProver-7B [8] | $2 \times 32 \times 600$ | 50.7% | – | 6/640 | – |
| | | $256 \times 32 \times 600$ | 65.9% | **27.0%** | – | – |
| | DeepSeek-Prover-V1.5-RL-7B + RMaxTS [10] | $4 \times 6400$ | 59.6% | 25.3% | – | – |
| | | $32 \times 6400$ | 63.5% | – | – | – |
| | HunyuanProver-7B [7] | $600 \times 8 \times 400$ | 68.4% | – | – | 20% |
| | BFS-Prover-7B [4] | $2048 \times 2 \times 600$ | 70.8% | – | – | **25%** |
| | | accumulative | 73.0% | – | – | – |
| Hybrid | DSP (GPT-4o, Isabelle) | 10 | – | – | 4/640 [55] | – |
| | DSP (Minerva-540B, Isabelle) [13] | 100 | 38.9% | – | – | 5% |
| | DSP+ (QwQ-32B, V3-671B, BFS-Prover-7B) | 1 | 52.5% | – | – | – |
| | (V3: shorthand for DeepSeek-V3-0324.) | 8 | 68.4% | – | – | – |
| | (R1: shorthand for DeepSeek-R1.) | 32 | 71.3% | 24.7% | 15/644 | – |
| | | 128 | 74.2% | **32.8%** | **24/644** | – |
| | | 1024 | 79.5% | – | – | **40%** |
| | DSP+ (QwQ-32B, QwQ-32B, BFS-Prover-7B) | 1024 | 79.1% | – | – | – |
| | DSP+ (R1-671B, V3-671B, BFS-Prover-7B) | 1024 | **80.7%** | – | – | – |
| | DSP+ (ensemble) | accumulative | **83.6%** | **33.9%** | **25/644** | **45%** |

## 6.1 The Synergy in DSP+

**Ablation of Components**. Figure 3 presents the results of different configurations, where one or more components are removed from the DSP+ framework. All variants exhibit performance degradation compared to the default "DSP+ (full)" setting. To further clarify each configuration, we take the "Draft + Sketch" setting as an example. In this setup, the Sketch model is directly prompted to output the final, complete Lean code directly given the natural language draft, which means it produces tactic-level code without any "sorry" placeholders. The remaining ablation settings are constructed in a similar manner. The "Sketch + Prove" setting shows a slight performance drop, benefiting from DeepSeek-V3's strong capability of one-step sketch generation. Interestingly, the "Draft + Sketch" configuration solves fewer problems than "DeepSeek-V3 only", highlighting the critical role of step provers within DSP+. Finally, the "BFS-Prover with Aesop" variant achieves much lower performance than reported in [4], primarily due to the constrained computational budget described in Section 5.1.

**Synergy from Ensemble Setting.** The synergy also manifests given the diversity of solved problems with different configurations. As listed in Table 2, we choose three configurations of draft and sketch models for pass@1024 and three others for pass@128, with the proving model fixed as BFS-Prover. The sample budgets are allocated according to their potential in toy experiments. And we find that different model combinations can cover different solved problems for the accumulative accuracy.

**Unique Proofs and Synergized Applications.** Besides the accuracy results, we also investigate the output of every phase. As discussed in Appendix M, the proofs generated by DSP+ differ from those by BFS-Prover for the same problems, with both improved readability and controllability. As a direct consequence of the synergy, DSP+ has facilitated us to analyze the unfinished subgoals after the proving phase and find eight wrongly formalized statements, which are detailed in Appendix F.

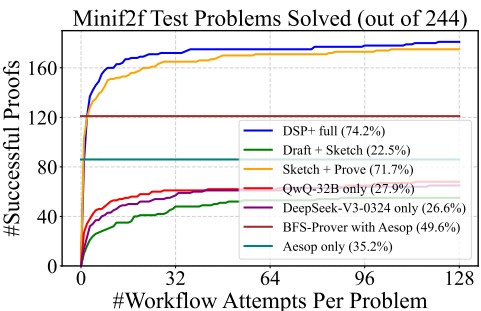

Figure 3: Ablation of DSP+ components, which shows the synergy in DSP+.

| Draft-Sketch, Pass@n | #Solved | #Accum. |
|---|---|---|
| **QwQ-V3@1024** | – | +194 |
| R1-V3@1024 | +5, -2 | +5 |
| QwQ-QwQ@1024 | +4, -5 | +2 |
| None-V3@128 | +1, -20 | +1 |
| QwQ(No format)-V3@128 | +1, -11 | +1 |
| R1-QwQ@128 | +3, -11 | +1 |
| | | Total: 204 |

Table 2: Configurations of DSP+ ensemble for miniF2F. (+x, -y) indicates the setting solves x new problems but with y unsolved w.r.t. the default setting. The accumulative solved problems (#Accum.) show the synergy of configurations.

## 6.2 The Effectiveness of Neuro-Symbolic Enhancements

**Conciseness of Draft.** We observe that removing the prompting for conciseness slightly improves performance with larger pass@k as shown in Figure 4. In fact, as in the case study of Appendix K, an unconstrained draft can be more informative with comparable length, which is acceptable for strong sketch models. Another interesting observation is that DSP+ shows higher accuracy without human informal proof, which is different from [13] and detailed in Figure 10 of Appendix D.

**Optimizations in Sketch.** In Figure 5, we conduct ablation studies on the two optimizations in the sketch phase: (1) the explicit specification of subgoal hypotheses, and (2) the error line masking. Removing either optimization leads to a significant drop in accuracy and sample efficiency.

**Symbolic Search with Proving Models.** This is studied in Figure 3 and Section 6.3.

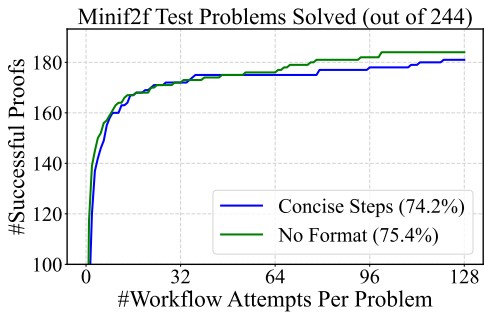

Figure 4: Ablation of Draft Formats. Free formatting is better as more informative.

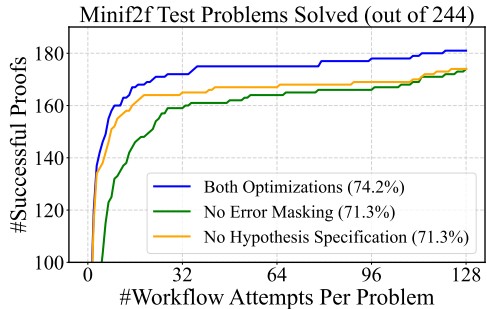

Figure 5: Ablation of Sketch Optimizations. Both optimizations are effective.

## 6.3 The Robustness of DSP+

**Different Draft Models.** As shown in Figure 6, DeepSeek-R1 achieves higher accuracy than QwQ-32B, indicating our default setting may not be optimal. We also observe that QwQ-32B exhibits stronger instruction following capability than DeepSeek-R1, which can bring higher sample efficiency (details in Appendix L). We also provide details about output tokens in Appendix D.

**Different Sketch Models.** As shown in Figure 7, we find QwQ-32B, despite being primarily designed for natural language reasoning, achieves results comparable to DeepSeek-V3, including solving some previously unsolved problems. This suggests the potential of QwQ-32B as the sketch model. We also provide more quantified metrics about them in Appendix D.

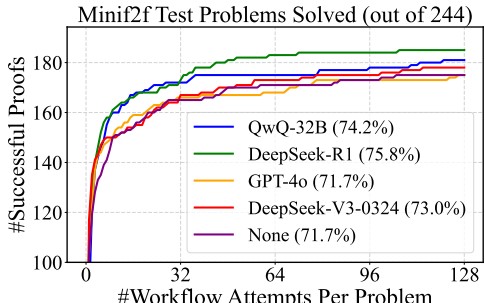

Figure 6: Comparing Different Draft Models. Reasoning models are optimal.

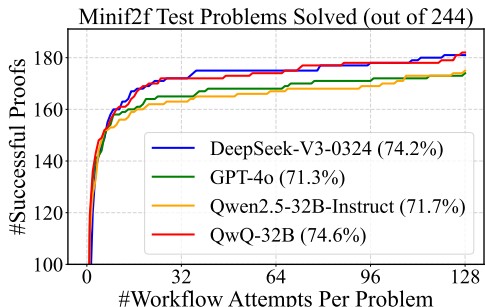

Figure 7: Comparing Different Sketch Models. DeepSeek-V3 and QwQ-32B are optimal.

**w/ and w/o Proving Models.** We study two configurations for the proving phase: the default setting with BFS-Prover and another with common tactics (Appendix A.1), which show pass@128 accuracy of 74.2% and 47.5%, respectively. We plan to integrate models like DeepSeek-Prover-V1.5-RL [10] and InternLM2.5-StepProver [52] in the future for more diversified capabilities (see Appendix I).

### 6.4 The Efficiency of DSP+

We collect the token statistics from [2] and [3]. As listed in Table 3. DSP+ can use fewer total inference tokens, which are #Average tokens per pass × #Passes, compared to Kimina-Prover-Preview, even for the same accuracy. And the most balanced configuration is our default configuration given the token efficiency and the dataset accuracy.

Table 3: Average inference cost of different solutions in miniF2F-test. DSP+ (R1-V3-BFS) can achieve the same accuracy with less total inference tokens w.r.t. Kimina-Prover Preview.

| Solution | Average tokens used per pass | | | | Accuracy |
| | 7B | 32B | 72B | 671B | |
|---|---|---|---|---|---|
| DSP+ (QwQ-V3-BFS) | 12k | 6.3k | – | 0.8k | 79.5 @ 1024 Pass |
| DSP+ (R1-V3-BFS) | 12k | – | – | 3.6k + 0.8k | 80.7 @ 1024 Pass |
| DSP+ (QwQ-QwQ-BFS) | 12k | 6.3k + 10k | – | – | 79.1 @ 1024 Pass |
| Kimina-Prover Preview [3] | – | – | 10k | – | 80.7 @ 8192 Pass |
| DeepSeek-Prover-V2 [2] | – | – | – | 6.75k | 88.9 @ 8192 Pass |

## 7 The Limitation of DSP+

**Underexplored Design Space**. Due to the vast design space of DSP+, we did not fully explore and set the optimal configuration as the default. And there are also opportunities to find the optimal configuration for the DSP ensemble given the differences in model capabilities and token efficiency. In addition, the DSP+ itself can be optimized to circumvent duplicated searches. All these can contribute to a more powerful and efficient search, which we leave as future work.

**Failure Cases.** We detail failure cases in Appendix E. In summary, we have identified challenges in every phase of DSP+, such as the proof of novelty for the draft, vague abstraction for the sketch, and misaligned difficulty for the proving. Interestingly, even different Lean 4 versions affect DSP+ accuracy. We regard both model training and better proof assistant support as essential to address these challenges, which require more computing resources and more expert effort, respectively.

## 8 Conclusion

In this work, we revisit the DSP framework, which resembles the human reasoning pattern from informal to formal. We find DSP is underestimated and can be revived as DSP+, which is as powerful as cutting-edge models for theorem proving. The key is our neuro-symbolic enhancement, which carefully coordinates reasoning models, symbolic search, and step provers in three phases.

With our comprehensive evaluations on benchmarks, we find DSP+ a universally applicable framework with high proving accuracy and token efficiency, which can potentially benefit the deployment

and the re-development from the theorem proving community. DSP+ can also serve in the pipeline for generating high-quality cold-start data for model training, as indicated in DeepSeekProver-V2.

We hope that, through our results and code, the community can have more diversified and synergized approaches for advanced theorem proving, besides the prevailing trend of large-scale training.

# 9 Acknowledgment

We thank the anonymous reviewers for their insightful feedback and constructive suggestions. We gratefully acknowledge the developers of the open-source BFS-Prover model, as well as the Qwen and DeepSeek model series, for providing invaluable resources supporting our research. All experiments in this work were conducted at Microsoft Research Asia.

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

# A Implementation Details

## A.1 The Configuration of Aesop

We configure Aesop's search space to be either tactics proposed by the step prover, or be with a few commonly used and efficient proof strategies (`rfl`, `linarith`, `nlinarith`, `ring`, `positivity`, `omega`, `ring_nf`, `ring_nf at *`, `simp`, `simp_all`, `field_simp`, `field_simp [*] at *`, `norm_num`, `norm_num [*] at *`, `norm_cast`, `norm_cast at *`). Furthermore, we modify Aesop's internal search prioritization to adopt the length-normalized scoring heuristic introduced in BFS-Prover [4].

In addition, the hypotheses specified by the sketch model are sometimes inaccurate. Therefore, during the proving phase, we split the budget into two strategies: one using only the hypotheses hinted with tactics (`clear * - ...`) (exemplified in Figure 2), and another using all available hypotheses. If either attempt succeeds, we consider the proof successful.

## A.2 The Motivation for the Built-in Integration of Step Provers

There are two common approaches to leverage the neural models in the proving phase: one is the state-to-theorem approach so that the converted theorems can be proved by models; the other is to externally parse and extract the state so that the step provers can help. However, both approaches are nontrivial with challenges, which motivate our built-in integration:

For the state-to-theorem approach, an example to demonstrate the challenge is shown in the figure below. The subgoal of `step4b` can be correctly proved within the original `test` theorem shown in the left. However, if the state corresponding to `step4b` is converted into a theorem, as shown in the right, the same proof fails due to the change of variable types. Similar issues are common—for instance, missing numbers or variable types often lead to incorrect converted theorems given Lean's foundation on the dependent type theory. We suspect the Lean state contains more information than what is visible at the string level. In VSCode's *InfoView* window, all numbers and variables can be queried for their specific types, but these details are not explicitly reflected in the string representation, leading to a loss of information. For parsing and extracting Lean proof state (e.g., using external Python), it is also challenging, as Lean code may involve complex tree structure, which is difficult to parse purely from code strings.

```
theorem test (m n : ℕ) (h₀ : m.gcd        -- Convert step 4b to theorem:
n = 6) (h₁ : m.lcm n = 126) : 60 ≤
m + n := by                               theorem step4b (m n : ℕ) (h₀ :
  let b := n / 6                          m.gcd n = 6) (h₁ : m.lcm n = 126)
  have step4b : n = 6 * b := by           (b : ℕ := n / 6): (n = 6 * b) :=
    /- State:                             by
      m n : ℕ                               /- State:
      h₀ : m.gcd n = 6                        m n : ℕ
      h₁ : m.lcm n = 126                      h₀ : m.gcd n = 6
      b : ℕ := n / 6                          h₁ : m.lcm n = 126
      ⊢ n = 6 * b                             b : optParam ℕ (n / 6)
    -/                                        ⊢ n = 6 * b
    rw [Nat.mul_div_cancel']                -/
    have h' := Nat.gcd_dvd_right m n        rw [Nat.mul_div_cancel']
    simp_all                                      tactic 'rewrite' failed
                                            have h' := Nat.gcd_dvd_right m n
                                            simp_all
```

Fortunately, at the Lean level—such as with `aesop`—it is possible to directly access Lean's internal structural information and run tactics based on a state, which can significantly improves efficiency.

Another independent approach in DeepSeekProver-V2 [2] demonstrates to reconstruct subgoals as theorems by purely syntactic parsing without referring to the Lean state, and therefore sidesteps the challenges we mention here. However, this approach can only work for the well-formatted sketch, not as generic and flexible as the built-in integration in DSP+.

### A.3 Lean Server Speedup

Even we have leveraged the built-in integration of tree search in Lean, which greatly reduces the need to recompile proven code, the verification process is still slow. The main timing bottleneck occurs during the header import (i.e., `import Mathlib`). Therefore, using the LeanREPL-based multi-process verification scheduler framework of DeepSeek-Prover-V1.5 [10], we support the reuse of the (`import Mathlib`) as a header, while still allowing different namespaces to be opened during verification. This results in a reduction of approximately 10 seconds in average for each verification.

### A.4 The Workflow of DSP+ and DSP+ Ensemble

Following Section 3, the entire DSP+ process with our default setting is as follows: a formal statement is first sent to QwQ-32B, which generates a draft. This draft is then passed to DeepSeek-V3-0324, which produces the initial sketch. Then, the sketch is processed with error line masking by interacting with the Lean environment via REPL [57]. Finally, the subgoals in the sketch are filled respectively with Aesop and BFS-Prover given the feedback from Lean. The DSP+ process terminates when the statement is proved, when resource parameters are exceeded, or when a timeout occurs. In this work, we try 6 combinations for DSP ensemble in miniF2F-test as listed in Table 2. And we only try R1-V3-BFS with pass@32 as the additional combination after default setting for ProofNet and PutnamBench.

## B  Sample Budget Configuration of Proving Phase

For the proving phase, we use $A \times W \times T$ for the maximum sample budget per subgoal, where $A$ denotes the number of search attempts, $W$ denotes the number of tactics generated for each expansion, $T$ denotes the number of expansion iterations, namely tree size. If not specified, $A = 8$, $W = 8$, and $T = 64$. However, few proving phases can actually use up its entire budget. The reasons are listed below:

**For parameter A.** Not every subgoal is difficult—many can be solved directly by symbolic search or resolved by a single BFS-Prover expansion.

**For parameter T.** In tree search, if no expandable nodes remain, the process terminates early. Since each node expands only to a width of 4 (by sampling 8 times and retaining at most 4 deduplicated expansions), this occurs frequently for hard problems.

**For timeout.** Additionally, a maximum verification time limit of 2400 seconds is set for each proving phase, resulting in the process often being terminated before the full budget is utilized.

We observe that under evaluation on miniF2F-test with Pass@32, each attempt of the DSP+ workflow samples the BFS-Prover for about 1500 times on average during the proving phase, and generated about 8 tokens per sample, far below the upper-bound $A \times W \times T$ per subgoal. This is well within acceptable limits for the relatively small BFS-Prover-7B model.

## C  Detailed Evaluation Results

Table 4 presents all recent top solutions on miniF2F. We also include the results of miniF2F subsets like miniF2F/IMO, miniF2F/AIME, and miniF2F/AMC in Table 5. The results are consistent with the overall trend observed in the main table.

As our independent interest, Table 6 presents DSP+ performance on ProverBench [2], which is introduced recently with little probability of data contamination. We can see DSP+ approaches the performance of DeepSeek-Prover-V2-671B, showing the generalization of our method.

## D  More Ablation Studies

**Detailed Ablation Studies of Draft Models.** Besides the accuracy of different models, we also investigate the average number of tokens in the generated proofs and the average number of tokens in the thinking process. The results are shown in Figure 8. We find that the QwQ model has a lower average number of answer tokens (AAT) and a higher average number of thinking tokens

Table 4: A collection of top solutions across three benchmarks (best results among the top 5 in bold). DSP+ and DSP+ ensemble are comparable to Kimina-Prover-Preview and DeepSeekProver-V2, outperforming all others.

| Type | Solution (Model Size) | Sample Budget | miniF2F-test | ProofNet-test | PutnamBench |
|---|---|---|---|---|---|
| Whole-proof Generation | InternLM2-StepProver-7B [31] | $1 \times 32 \times 100$ | 48.8% | 18.1%[*] | 4/640 [55] |
| | Leanabell-Prover-7B [44] | 128 | 61.1% | – | – |
| | DeepSeek-Prover-V1.5-RL-7B [10] | $4 \times 6400$ | 58.4% | 23.7% | – |
| | | $16 \times 6400$ | 60.2% | – | – |
| | Goedel-Prover-7B [5] | 32 | 57.6% | 15.2% | 6/644 |
| | | 512 | 62.7% | – | 7/644 |
| | | 25600 | 64.7% | – | – |
| | STP-7B [6] | 3200 | 65.0% | 23.9% | 8/644 |
| | | 25600 | 67.6% | 26.9% | – |
| | Kimina-Prover-Preview-7B [3] | 1 | 52.5% | – | – |
| | | 32 | 63.1% | – | – |
| | | 192 | – | – | **10/644 [55]** |
| | | 1024 | 70.8% | – | – |
| | Kimina-Prover-Preview-72B [3] | 1 | 52.9% | – | – |
| | | 8 | 65.2% | – | – |
| | | 32 | 68.9% | – | – |
| | | 1024 | 77.9% | – | – |
| | | 8192 | **80.7%** | – | – |
| | DeepSeek-Prover-V2-7B [2] | 1 | 58.6% | – | – |
| | | 32 | 75.6% | 23.0% | 11/658 |
| | | 128 | – | 25.4% | 15/658 |
| | | 1024 | 79.9% | **29.6%** | **23/658** |
| | | 8192 | **82.0%** | – | – |
| | DeepSeek-Prover-V2-671B [2] | 1 | 61.9% | – | – |
| | | 32 | 82.4% | 30.5% | 22/658 |
| | | 128 | – | 33.6% | 33/658 |
| | | 1024 | 86.6% | **37.1%** | **49/658** |
| | | 8192 | **88.9%** | – | – |
| | OpenAI o3-mini | 32 | 24.6% [3] | – | – |
| | gemini-2.5-pro-preview-03-25 | 32 | 37.7% [3] | – | – |
| | ReProver-229M [33] | – | 26.5% | 13.8%[*] | 0/640 |
| | GPT-4o | 10 | – | – | 1/644 [55] |
| | DeepSeek-R1-671B | 1 | – | – | 1/644 [55] |
| | DeepSeek-V3-0324-671B | 1 | – | – | 0/644 [55] |
| | | 32 | 25.0% | | |
| Tree Search | InternLM2.5-StepProver-7B [8] | $2 \times 32 \times 600$ | 50.7% | – | 6/640 |
| | | $256 \times 32 \times 600$ | 65.9% | 27.0%[*] | – |
| | DeepSeek-Prover-V1.5-RL-7B + RMaxTS [10] | $4 \times 6400$ | 59.6% | 25.3% | – |
| | | $32 \times 6400$ | 63.5% | – | – |
| | HunyuanProver-7B [7] | $600 \times 8 \times 400$ | 68.4% | – | – |
| | BFS-Prover-7B [4] | $2048 \times 2 \times 600$ | 70.8% | – | – |
| | | accumulative | 73.0% | – | – |
| | ABEL-8B [9] | 596 | – | – | 7/640 |
| | | $1 \times 128 \times 64$ | 41.3% | – | – |
| Hybrid | DSP (GPT-4o, Isabelle) | 10 | – | – | 4/640 [55] |
| | DSP (Minerva-540B, Isabelle) [13] | 100 | 38.9% | – | – |
| | DSP+ (QwQ-32B, V3-671B, BFS-Prover-7B) | 1 | 52.5% | – | – |
| | (V3: shorthand for DeepSeek-V3-0324.) | 8 | 68.4% | – | – |
| | (R1: shorthand for DeepSeek-R1.) | 32 | 71.3% | 24.7% | 15/644 |
| | | 128 | 74.2% | **32.8%** | **24/644** |
| | | 1024 | 79.5% | – | – |
| | DSP+ (QwQ-32B, QwQ-32B, BFS-Prover-7B) | 1024 | 79.1% | – | – |
| | DSP+ (R1-671B, V3-671B, BFS-Prover-7B) | 1024 | **80.7%** | – | – |
| | DSP+ (ensemble) | accumulative | **83.6%** | **33.9%** | **25/644** |

[*] ProofNet-all results are used due to unavailable ProofNet-test data.

Table 5: Performance of top solutions on IMO, AIME and AMC problems of miniF2F-test. DSP+ and DSP+ ensemble are comparable to Kimina-Prover-Preview and DeepSeekProver-V2.

| Solution | Sample budget | miniF2F-test | miniF2F/IMO | miniF2F/AIME | miniF2F/AMC |
|---|---|---|---|---|---|
| Hunyuan-Prover-7B | $600 \times 8 \times 400$ | 68.4% | 20.0% | – | – |
| BFS-Prover-7B | $2048 \times 2 \times 600$ | 70.8% | 25.0% | – | – |
| Kimina-Prover-Preview-72B | 8192 | 80.7% | 40.0% | 86.7% | 66.7% |
| DeepSeek-Prover-V2-671B | 8192 | 88.9% | 50.0% | 93.3% | 77.8% |
| DSP+ (QwQ-V3-BFS) | 1024 | 79.5% | 40.0% | 86.7% | 64.4% |
| DSP+ (ensemble) | accumulative | 83.6% | 45.0% | 86.7% | 71.1% |

Table 6: Performance of top solutions on ProverBench. DSP+ and DSP+ ensemble are comparable to DeepSeekProver-V2.

| Solution | Sample Budget | ProverBench |
|---|---|---|
| STP-7B | 32 | 27.5% |
| | 128 | 31.4% |
| | 512 | 36.3% |
| DeepSeek-Prover-V2-7B | 32 | 49.0% |
| | 128 | 50.8% |
| | 512 | 51.7% |
| DeepSeek-Prover-V2-671B | 32 | 52.9% |
| | 128 | 56.5% |
| | 512 | 59.1% |
| DSP+ (QwQ-V3-BFS) | 32 | 46.77% |
| | 128 | **52.92%** |
| DSP+ (ensemble) | accumulative | **55.69%** |

(ATT) compared to DeepSeek-R1. Interestingly, reasoning models show a lower AAT compared to non-reasoning models, which demonstrate that the thinking tokens can benefit the conciseness of the draft.

**Detailed Ablation Studies of Sketch Models.** Besides the accuracy, we also investigate the average translation rate (ATR) and median translation rate (MTR) of different sketch models. The results are shown in Figure 9. We find that the DeepSeek-V3-0324 model has a higher ATR and MTR compared to other models, indicating that it is more effective in generating correct code lines. This suggests that the DeepSeek-V3-0324 model is better suited for the sketch phase of DSP+.

**Ablation Studies of Human Informal Proof.** We also investigate whether human-written proofs help improve accuracy, by providing them to the draft and sketch phases respectively, and comparing the results with counterparts that do not include human proofs as shown in Figure 10. We find that incorporating informal proofs does not lead to an overall improvement in performance, which is different from the original DSP [13].

# E   Failure Cases of DSP+

We observe a wide range of failure modes, with most attributed to the following categories.

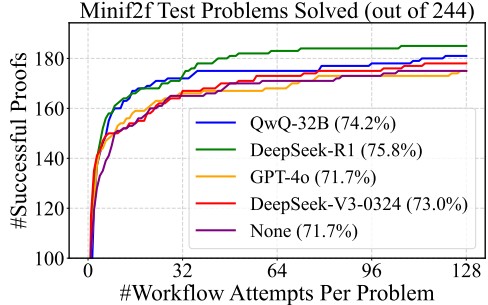

| Model | AAT[a] | ATT[b] |
|---|---|---|
| **QwQ-32B** | 575 | 5682 |
| DeepSeek-R1 | 693 | 2888 |
| GPT-4o-2024-11-20 | 748 | – |
| DeepSeek-V3-0324 | 948 | – |
| None | – | – |

[a]AAT: Average Answer Token
[b]ATT: Average Thinking Token

Figure 8: The Performance of DSP+ with Different Draft Models. Reasoning models are better than non-reasoning models with the help of the thinking tokens.

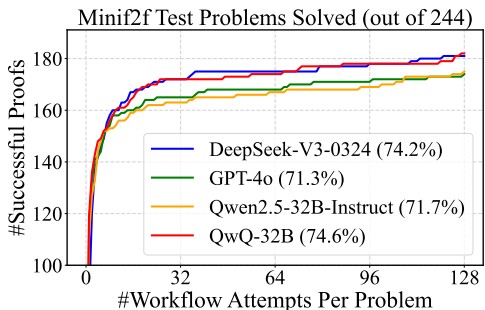

| Model | ATR[a] | MTR[b] |
|---|---|---|
| **DeepSeek-V3-0324** | **84.2%** | **100.0%** |
| QwQ-32B | 63.2% | 70.0% |
| Qwen2.5-32B-Instruct | 71.0% | 80.0% |
| GPT-4o-2024-11-20 | 77.6% | 89.2% |

[a]ATR: Average Translation Rate. During the Sketch phase, erroneous code lines are iteratively removed until the code is syntactically correct. The translation rate is defined as the percentage of code lines (excluding headers, comments, and blank lines) that remains after removal.
[b]MTR: Median Translation Rate

Figure 9: The Performance of DSP+ with Different Sketch Models. DeepSeek-V3 shows strong capability of autoformalization with high translation rate.

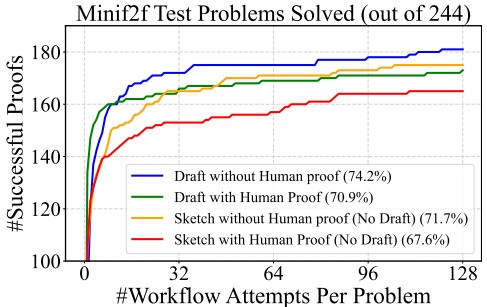

Figure 10: Ablation of Human Informal Proof. Human informal proof offers no help for reasoning models to generate good draft in miniF2F.

## E.1 Limitations in Draft Phase

Many IMO problems involve very long natural language proofs, such as `imo_1977_p6` (Reference Solution: `https://artofproblemsolving.com/wiki/index.php/1977_IMO_Problems/Problem_6`). These problems and their solutions are challenging even for humans to come up with, and thus the same for the reasoning models. Once the draft is incorrect for the intuition, DSP+ workflow struggles to produce a correct proof, because the sketch and proving phases are faithful translators and executors. Here is an example of unsuccessful attempt:

---

**Wrong Draft**

**Informal Statement:** Let $f(n)$ be a function $f : \mathbb{N}^+ \to \mathbb{N}^+$. Prove that if $f(n+1) > f(f(n))$ for each positive integer $n$, then $f(n) = n$
<think>...</think>
**Step 1:** $f(1) = 1$
**Step 2:** Assume $f(k) = k$ for all $k \leq n$
**Step 3:** $f(n) = n$
**Step 4:** $f(f(n)) = f(n) = n$
**Step 5:** $n < f(n+1)$
**Step 6:** $f(n+1) \geq n+1$
**Step 7:** Suppose $f(n+1) = n+1+k$ with $k \geq 1$
**Step 8:** $f(n+1+k) < f(n+2)$
**Step 9:** $n+1+k \geq n+2$
**Step 10:** $f(n+1+k) \geq f(n+2)$
**Step 11:** Contradiction implies $k = 0$
**Step 12:** $f(n+1) = n+1$
**Step 13:** $f(n) = n$ for all $n \in \mathbb{N}$

---

### E.2 Limitations in Sketch Phase

**Difficulty of Formalization.** Some solutions are inherently difficult to formalize in Lean. For instance, the standard solution to `amc12a_2020_p9` (Reference Solution: `https://artofproblemsolving.com/wiki/index.php/2020_AMC_12A_Problems/Problem_9`) requires the use of the function graph to solve the problem, which could be hard to interpret in Lean. Moreover, since this is originally a multiple-choice problem, even the standard solution lacks full formal rigor, making it still some distance away from being accepted in Lean.

> **AoPS's Solution to 2020 AMC 12A Problem 9**
>
> 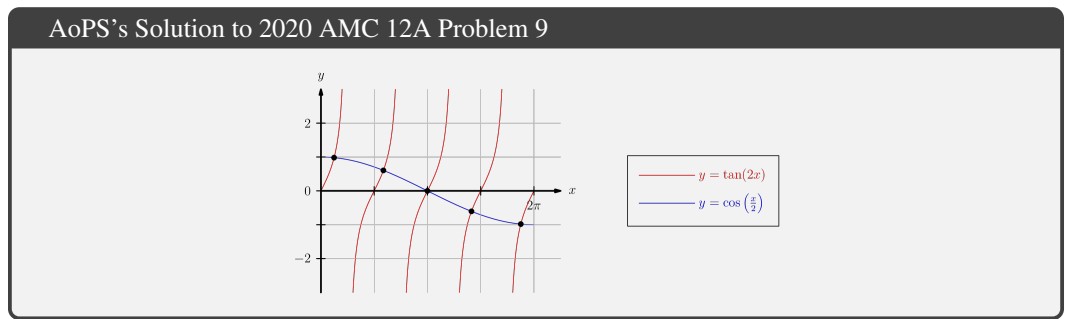

**Challenge of Lean's Syntax.** Lean's syntax requires much higher level of rigor and delicateness than natural language, and it can sometimes produce unexpected errors. This poses a serious challenge for the sketch model, which does not have access to real-time Lean feedback. For example, the type of a number is often inferred from the context, and if the context is unclear or inconsistent, the system may raise obscure type errors. Below, we present an example to illustrate such a challenging issue:

```
import Mathlib
example (x : ℝ) (h₀: x ^ 2 = 1) : x = 1 ∨ x = -1 := by exact
sq_eq_one_iff.mp h₀
example (x : ℝ) (h₀: 2 ^ x = 1) : x = 0 := by sorry
```

In Lean, the first expression compiles correctly, while the second one raises the error: `failed to synthesize HPow ℕ ℝ ?m.xxx`, because 2 is treated as a natural number, and the operation of automatically converting a natural number to a real power is undefined. These nuances are not convenient to humans, which also applies to LLM.

**Misalignment of Function Definitions.** The definitions of functions in Lean does not always align with human intuition, especially those learned from the standard curriculum. As mentioned in the Appendix F, the domain of the `Real.log()` function in Lean is all real numbers. Similarly, functions defined on the entire real number domain, such as `HDiv.hDiv`, `Real.sqrt`, `Real.arcsin`, etc., may have values outside their domains that appear illegal to humans, yet they can still significantly impact the proof of theorems. The above findings can be demonstrated with the following code, which holds true in `Lean 4:v4.17.0-rc1`:

```
import Mathlib
example : 1 / 0 = 0 := by simp
example : 1 - 2 = 0 := by simp
example : Real.log 0 = 0 := by simp
example : Real.arcsin 2 = Real.pi / 2 := by simp
example : Real.sqrt (-1) = 0 := by simp [Real.sqrt]
```

That is to say, while $1 - 2 = -1$ holds in the integers, the value $-1$ does not exist in the natural number domain. As a result, in Lean's definition over natural numbers, we have $1 - 2 = 0$. This means that subtraction on natural numbers often fails to satisfy the commutative law.

### E.3 Limitations in Proving Phase

Some statements may appear trivial in natural language but are counter-intuitively difficult to verify formally. For example, showing that $\pi > 3.1415$, proving that the positive divisors of $81$ are exactly $\{1, 3, 9, 27, 81\}$, or deducing $a = 7$, $b = 11$ from the equation $a + \sqrt{b} = 7 + \sqrt{11}$ when $a$ and $b$ are integers, all seem straightforward, yet constructing a rigorous proof can be unexpectedly challenging.

```
import Mathlib
example : Real.pi > 3.141 := by sorry
example (x y : ℕ) (h: x * y = 81): x ∈ ({1,3,9,27,81} : Finset ℕ) := by
sorry
example (a b : ℤ) (h: a + Real.sqrt b = 7 + Real.sqrt 11): a = 7 ∧ b = 11 :=
 by sorry
```

### E.4  The Impact of Lean Versions

We use the miniF2F dataset from DeepSeek-Prover-V1.5 [11], which employs `Lean 4:v4.9.0-rc1`. Our experiments reveal that the `rfl` tactic can be used to solve some problems in this version, but the same proofs fail in `Lean 4:v4.17.0-rc1`, suggesting behavioral differences in the tactic's implementation across versions. Below is an example that is affected by the Lean version:

```
open BigOperators Real Nat Topology Rat
theorem mathd_numbertheory_233 (b : ZMod (11 ^ 2)) (h₀ : b = 24⁻¹) : b = 116
:= by
  subst h₀
  simp_all only [reducePow]
  rfl
```

## F  Identified Errors in the miniF2F-test Dataset

With DSP+, we identify 8 incorrect problems in the miniF2F-test dataset[5] The discovered problems are: `amc12a_2020_p7`, `amc12a_2020_p10`, `amc12a_2021_p9`, `imo_1968_p5_1`, `induction_prod1p1onk3le3m1onn`, `mathd_algebra_158`, `mathd_algebra_342`, `mathd_numbertheory_343`. These problems are either unprovable or contain translation errors, causing a mismatch between the Lean formalization and the natural language description.

**Example:** `amc12a_2020_p10`

The original Lean theorem is stated as:

```
theorem amc12a_2020_p10 (n : ℕ) (h₀ : 0 < n) (h₁ : Real.logb 2 (Real.logb 16
n) = Real.logb 4 (Real.logb 4 n)) : (List.sum (Nat.digits 10 n)) = 13 := by
```

Since our workflow allows partial proofs containing `sorry`, we find that 2 out of the 12 generated subgoals could not be proven in one attempt. The details of the proof attempt are shown below.

```
theorem amc12a_2020_p10 (n : ℕ) (h₀ : 0 < n) (h₁ : Real.logb 2 (Real.logb 16
n) = Real.logb 4 (Real.logb 4 n)) : (List.sum (Nat.digits 10 n)) = 13 := by
...
  -- Step 3: Substitute into original equation
  have step3 : Real.logb 2 ((Real.logb 4 n) / 2) = Real.logb 4 (Real.logb 4
n) := by ...
  -- Step 4: Split log of quotient
  have step4 : Real.logb 2 (Real.logb 4 n) - Real.logb 2 2 = Real.logb 4
(Real.logb 4 n) := by
    sorry
...
  -- Step 8: Solve for log₂(log₄n)
  have step8 : Real.logb 2 (Real.logb 4 n) = 2 := by ...
  -- Step 9: Exponentiate to solve for log₄n
  have step9 : Real.logb 4 n = 4 := by
    sorry
...
```

---

[5]Kimina-Prover Preview project identifies 5 incorrect problems in miniF2F-test, 2 of which—`aime_1994_p3` and `mathd_numbertheory_618`—are not discovered by us. We use their corrected versions in our experiments.

From Step 3 to Step 4, the LLM applies the logarithm quotient rule, i.e.,

$$\log_b(x/y) = \log_b x - \log_b y.$$

We can use tools such as loogle, mathlib4 docs, or leansearch to find how this theorem is defined in Lean. In Lean, it is stated as:

```
theorem Real.logb_div (hx : x ≠ 0) (hy : y ≠ 0) : logb b (x / y) = logb b x
  - logb b y
```

Therefore, if we want to apply the logarithm quotient rule, we also need to prove that `Real.logb 4 n ≠ 0`. One might assume that since `Real.logb 4 n` appears as the argument of a logarithm in the original problem, it must be nonzero. Unfortunately, this cannot be proven in Lean, because in Lean, the definition of `Real.logb` is:

> As with the natural logarithm, we define `logb b x` to be `logb b |x|` for $x < 0$, and 0 for $x = 0$.

In this definition, `Real.logb 4 n` can be zero even though it appears as the argument of another `Real.logb`. We now verify whether it is provable that `Real.logb 4 n ≠ 0` under the assumption that $n > 0$.

Unfortunately, the whole equation still holds when $n = 1$, which can be verified in Lean with:

```
theorem amc12a_2020_p10 (n : ℕ) (h₀ : n = 1) : Real.logb 2 (Real.logb 16 n) =
  Real.logb 4 (Real.logb 4 n) := by simp_all
```

So we are unable to establish that `Real.logb 4 n ≠ 0`, which prevents the application of the logarithm quotient rule in Step 4.

Furthermore, Step 8 to Step 9 also presents difficulties. Although it may appear straightforward for humans, `Real.logb 4 n` could potentially be $-4$, and this case must be explicitly ruled out to ensure correctness.

In our revised version of the `miniF2F` dataset, we modify the condition $(h_0 : 1 < n)$ to ensure that all arguments passed to `Real.logb` are strictly positive, consistent with the domain of the logarithm function over real numbers as understood by humans. We also make adjustments to some other problems with domain definitions inconsistent with human interpretations, although in some cases, the parts outside the commonly accepted domain could be ruled out by contradiction.

# G  Prompts Used in This Work

---

**Prompt for Draft Model**

formal_statement:
{formal_statement}
Please provide an extremely detailed mathematical calculation following your thinking. Each step can only contain **one** equation without any explanation.
Here is an example:
### Step 1:
\[ x + y + xy = 80 \]
. . .
### Step 5:
\[ x + y + xy + 1 = 81 \]

---

**Prompt for Sketch Model**

informal_proof:
{detailed_informal_proof}
Prove the theorem in Lean 4 code. You should translate steps in the informal proof in a series of 'have'/'let'/'induction'/'match'/'suffices' statements, but you do not need to prove them. You only need to use placeholder 'by{{new_line}}prove_with[h1, step5, . . .{{hypothesises used here which are proposed ahead}}]'. We want to have as many lemmas as possible, and every lemma must be easy to proof.

When using a / b, you must specify **a's or b's type**, because (1:ℝ) / 2 is 0.5, but (1:ℤ) / 2 is 0.
When using a − b, you must specify **a's or b's type**, because (1:ℤ) − 2 is −1, but (1:ℕ) − 2 is 0.
n! is incorrect, you should use (n)!.
Here is an example:

```
import Mathlib
example (x y : ℝ) (h1 : x ≤ 1 / 2) (h2 : x > 0) (t: y < Real.sin
(x)): y < 1 / 2 := by
  -- Step 1
  have h3 : y < (1:ℝ) / 2 := by
    -- Step 2
    have h4 : Real.sin x ≤ x := by
      prove_with[h2]
    -- Step 3
    have h5 : y < x := by
      prove_with[h4, t]
    prove_with[h1, h5]
  exact h3
```

formal_statement:
```lean4
{header}
{formal_statement}
```

# H Illustration of DSP+ Workflow with an IMO Problem

Our method successfully proves `imo_2019_p1`, whereas all prior arts have failed to solve this problem. Here we present its entire trace. However, due to the extensive length of QwQ's reasoning, we are unable to display the thinking part of draft phase here. In this problem, we can find all the techniques introduced by our fine-grained and integrated neuro-symbolic enhancements. To clarify the effect of enhancements, we use *Comment* to highlight our observations, which is not generated by the sketch model.

---

**Formal Statement**

```
open BigOperators Real Nat Topology Rat
theorem imo_2019_p1 (f : ℤ → ℤ) :
  (∀ a b, f (2 * a) + 2 * f b = f (f (a + b))) ↔ ∀ z, f z = 0 ∨ ∃ c, ∀ z,
f z = 2 * z + c := by
```

---

**Draft**

<think>...</think>
**Step 1**: $f(2a) + 2f(b) = f(f(a+b))$  $\forall a, b \in \mathbb{Z}$
**Step 2**: $f(0) + 2f(b) = f(f(b))$  $\forall b \in \mathbb{Z}$
**Step 3**: $f(2a) + 2f(0) = f(f(a))$  $\forall a \in \mathbb{Z}$
**Step 4**: $f(0) + 2f(a) = f(2a) + 2f(0)$  $\forall a \in \mathbb{Z}$
**Step 5**: $f(2a) = 2f(a) - f(0)$  $\forall a \in \mathbb{Z}$
**Step 6**: $f(2) + 2f(b) = f(f(b+1))$  $\forall b \in \mathbb{Z}$
**Step 7**: $f(f(b+1)) = f(0) + 2f(b+1)$  $\forall b \in \mathbb{Z}$
**Step 8**: $f(2) + 2f(b) = f(0) + 2f(b+1)$  $\forall b \in \mathbb{Z}$
**Step 9**: $2f(b+1) = f(2) - f(0) + 2f(b)$  $\forall b \in \mathbb{Z}$
**Step 10**: $f(b+1) = \frac{f(2)-f(0)}{2} + f(b)$  $\forall b \in \mathbb{Z}$
**Step 11**: $f(b+1) - f(b) = \frac{f(2)-f(0)}{2}$  $\forall b \in \mathbb{Z}$
**Step 12**: $f(z) = mz + c$  where $m = \frac{f(2)-f(0)}{2}$ and $c = f(0)$
**Step 13**: $2m(a+b) + 3c = m^2(a+b) + (m+1)c$  $\forall a, b \in \mathbb{Z}$
**Step 14**: $m^2 - 2m = 0 \Rightarrow m(m-2) = 0$
**Step 15**: $2c = (m+1)c \Rightarrow (2-m)c = 0$
**Step 16**: $m = 0 \Rightarrow c = 0$ or $m = 2$ and $c$ is arbitrary
**Step 17**: $f(z) = 0$ or $f(z) = 2z + c$  $\forall z \in \mathbb{Z}$

**Corresponding draft steps in comment, sketch on the red left, proof on the green right**

```
open BigOperators Real Nat Topology Rat
theorem imo_2019_p1 (f : ℤ → ℤ) : (∀ a b, f (2 *
a) + 2 * f b = f (f (a + b))) ↔ ∀ z, f z = 0 ∨ ∃
c, ∀ z, f z = 2 * z + c := by
  constructor
  · -- Forward direction: assume equation holds,
show f is 0 or linear
    intro h
    --  Step 2
    have h2 : ∀ b, f 0 + 2 * f b = f (f b) := by
    intro b
```
```
    -- exact h (0 : ℤ) b        have h1 := h 0 b
    sorry                       simp_all
```
```
    -- Step 3
    have h3 : ∀ a, f (2 * a) + 2 * f 0 = f (f a) := by
    intro a
```
```
    -- exact h a (0 : ℤ)        (simp_all)
    sorry
```
```
    -- Step 5
    have h4 : ∀ b, f (2 * b) = 2 * f b - f 0 := by
```
```
    prove_with [h2, h3]         clear * - h2 h3
                                intro b
                                have h := h3 b
                                linarith [h2 b, h3 b]
```
```
    -- *Comment*:  h5 is missing given LLM's
unexplainable interpretation.
    -- Step 6
    have h6 : ∀ b, f 2 + 2 * f b = f (f (b + 1)) := by
    intro b
```
```
    -- exact h (1 : ℤ) b        have h5 := h 1 b
    sorry                       rw [add_comm] at h5
                                rw [add_comm] at h5
                                rw [add_comm] at h5
    -- *Comment*: Tree         rw [add_comm] at h5
search may lead to             rw [add_comm] at h5
looped transitions            rw [add_comm] at h5
between two states. We         rw [add_comm] at h5
will address this issue        rw [add_comm] at h5
in the future.                 rw [add_comm] at h5
                               simpa [add_comm,
                               mul_comm] using h5
```
```
    -- Step 7
    have h7 : ∀ b, f (f (b + 1)) = f 0 + 2 * f (b +
1) := by
    intro b
```
```
    -- exact h2 (b + 1)         (simp_all)
    sorry
```
```
    -- Step 8
    have h8 : ∀ b, f 2 + 2 * f b = f 0 + 2 * f (b +
1) := by
```
```
    prove_with [h6, h7]         clear * - h6 h7
                                (simp_all)
```
```
    -- Step 9
    have h9 : ∀ b, 2 * f (b + 1) = (f 2 - f 0) + 2 *
f b := by
```
```
    prove_with [h8]             clear * - h8
                                exact fun b ↦ by linarith
                                [h8 b]
```
```
    -- Step 10
    have h10 : ∀ b, f (b + 1) = ((f 2 - f 0) / (2 : ℤ
)) + f b := by
```
```
    prove_with [h9]             clear * - h9
                                intro b
                                have h10 := h9 (b + 1)
                                have h11 := h9 (b + 1)
                                have h12 := h9 b
                                omega
```

```
    -- Step 11
    have h11 : ∀ b, f (b + 1) - f b = (f 2 - f 0) /
(2 : ℤ) := by
```
```
    prove_with [h10]            clear * - h10
                                (simp_all)
```
```
    let m : ℤ := (f 2 - f 0) / (2 : ℤ)
    let c : ℤ := f 0
    -- Step 12
    have h12 : ∀ z, f z = m * z + c := by
```
```
    prove_with [h10]            clear * - h10
                                intro z
                                induction' z using
                                Int.induction_on with z ih z ih
                                · ring
                                · rw [h10]
                                  linarith
                                · have h1 := h10 (-z)
                                  have h2 := h10 (-z - 1)
                                  simp_all
                                  linarith
```
```
    -- Step 13
    have h13 : ∀ a b, (m * (2 * a) + c) + 2 * (m * b +
c) = m * (m * (a + b) + c) + c := by
```
```
    prove_with [h12]            (simp_all)
```
```
    -- Step 14
    have h14 : m * (m - 2) = 0 := by
```
```
    prove_with [h13]            clear * - h13
                                have h14 := h13 1 0
                                ring_nf at h14 ⊢
                                linarith [h13 2 0, h14]
```
```
    -- Step 15
    have h15 : (2 - m) * c = 0 := by
```
```
    prove_with [h13]            clear * - h13
                                have h14 := h13 1 0
                                ring_nf at h14 ⊢
                                linarith [h13 0 2, h14]
```
```
    -- Step 16
    have h_m_eq0_or_2 : m = 0 ∨ m = 2 := by
```
```
    prove_with [h14]            clear * - h14
                                (simp_all)
                                (omega)
```
```
    cases h_m_eq0_or_2 with -- Step 16
    | inl h_m0 =>
    have h_c0 : c = 0 := by
```
```
    prove_with               clear * - h15 h_m0
[h15, h_m0]               (nlinarith)
```
```
    have h_f_zero : ∀ z, f z = 0 := by
```
```
    prove_with               clear * - h12 h_m0 h_c0
[h12, h_m0, h_c0]         (simp_all)
```
```
    exact fun z => Or.inl (h_f_zero z)
    | inr h_m2 =>
    have h_exists_c : ∃ c', ∀ z, f z = 2 * z + c' :=
by
    use c
    intro z
    have eq : f z = m * z + c := h12 z
    rw [h_m2] at eq
    exact eq
```
```
    -- exact fun z =>          (simp_all)
Or.inr ⟨c, h_exists_c⟩
    sorry
```

```
· -- Reverse direction: assume f is 0 or linear, show equation holds
 intro h_rev
 have h_rev_split : (∀ z, f z = 0) ∨ (∃ c, ∀ z, f z = 2 * z + c) := by
```

```
  prove_with [h_rev]                                    clear * - h_rev
--============================================          by_cases h : ∃ c : ℤ, ∀ z : ℤ, f z = 2 * z + c
  -- cases h_rev_split with                             · right
  -- | inl h_all_zero =>                                  exact h
  --   intro a b                                        · left
  --   simp [h_all_zero, f]                               exact fun z ↦ (h_rev z).resolve_right h
  -- | inr ⟨c, h_f_eq⟩ =>                              --============================================
  --   intro a b                                        intro a b
  --   have lhs_val : f (2 * a) + 2 * f b = 4 * (a +    cases h_rev_split with
b) + 3 * c := by                                        | inl h => (simp_all)
  --     rw [h_f_eq (2 * a), h_f_eq b]                  | inr h_1 =>
  --     ring_nf                                          obtain ⟨w, h⟩ := h_1
  --   have rhs_val : f (f (a + b)) = 4 * (a + b) +      (norm_num [*] at *)
3 * c := by                                              (linarith)
  --     rw [h_f_eq (a + b), h_f_eq (2 * (a + b) +
c)]                                                    -- *Comment*: The left part has some errors, and
  --     ring_nf                                        the Error Line Masking technique successfully
  --   rw [lhs_val, rhs_val]                            replaces the corresponding code with tactic sorry.
  sorry                                                 This is proved later by BFS-Prover.
```

# I  Capability Comparison of DeepSeek-Prover-V1.5-RL and BFS-Prover

Although BFS-Prover is a powerful prover, we use it more because its input is solely state, which fits our framework. This offers benefit given InternLM2.5-StepProver and DeepSeekProver-v1.5 both require chain-of-thought prefix, which is incompatible to our current framework. However, in terms of performance, BFS-Prover cannot outpace all current provers. Here is an example:

```
example (b : ℝ) (step6 : b ^ 9 = (2 / 3) ^ 9): (b = 2 / 3) := by
  apply Eq.symm
  nlinarith [step6, sq_nonneg (b ^ 2 - (2 / 3) ^ 2), sq_nonneg (b ^ 3 - (2 /
3) ^ 3), sq_nonneg (b ^ 4 - (2 / 3) ^ 4), sq_nonneg (b ^ 5 - (2 / 3) ^ 5),
sq_nonneg (b ^ 6 - (2 / 3) ^ 6), sq_nonneg (b ^ 7 - (2 / 3) ^ 7), sq_nonneg
(b ^ 8 - (2 / 3) ^ 8), sq_nonneg (b ^ 9 - (2 / 3) ^ 9)]
```

This statement appears as a subgoal. Unfortunately, under our configuration, BFS-Prover cannot solve it. By contrast, we find DeepSeek-Prover-V1.5-RL can provide the correct proof given moderate search budget. This indicates the diversified capabilities of different models.

# J  Success Cases of DSP+

## J.1  Using Jensen's Inequality to Solve an Inequality

### Statement

```
open BigOperators Real Nat Topology Rat
theorem algebra_apbon2pownleqapownpbpowon2 (a b : ℝ) (n : ℕ) (h₀ : 0 < a ∧
0 < b) (h₁ : 0 < n) : ((a + b) / 2) ^ n ≤ (a ^ n + b ^ n) / 2 := by
```

### Draft

<think>...</think>
**Step 1:** $\left(\frac{a+b}{2}\right)^n \leq \frac{a^n+b^n}{2}$
**Step 2:** Let $f(x) = x^n$ for $x > 0$ and $n \in \mathbb{N}$
**Step 3:** Since $f''(x) = n(n-1)x^{n-2} \geq 0$ for $x > 0$, $f$ is convex.
**Step 4:** By Jensen's inequality for convex functions: $f\left(\frac{a+b}{2}\right) \leq \frac{f(a)+f(b)}{2}$
**Step 5:** $\left(\frac{a+b}{2}\right)^n \leq \frac{a^n+b^n}{2}$
**Step 6:** Alternatively, consider using induction...
...

## Sketch

```
open BigOperators Real Nat Topology Rat
theorem algebra_apbon2pownleqapownpbpowon2 (a b : ℝ) (n : ℕ) (h₀ : 0 < a ∧
0 < b) (h₁ : 0 < n) : ((a + b) / 2) ^ n ≤ (a ^ n + b ^ n) / 2 := by
  -- Step 1: Define the function f(x) = x^n
  let f : ℝ → ℝ := fun x => x ^ n
  -- Step 2: Show that f is convex
  have h2 : ConvexOn ℝ (Set.Ioi 0) f := by
    prove_with[]
  -- Step 3: Apply Jensen's inequality to f
  have h3 : f ((a + b)/2) ≤ (f a + f b)/2 := by
    have h4 : a ∈ Set.Ioi 0 ∧ b ∈ Set.Ioi 0 := by
      prove_with[h₀]
    have h5 : (a + b)/2 ∈ Set.Ioi 0 := by
      prove_with[h₀]
    have h6 : (1/2 : ℝ) + (1/2 : ℝ) = 1 := by
      prove_with[]
    prove_with[h2, h4, h5, h6]
  -- Step 4: Rewrite using definition of f
  have h7 : ((a + b)/2) ^ n ≤ (a ^ n + b ^ n)/2 := by
    prove_with[h3]
  -- Final conclusion
  exact h7
```

## Proof

```
theorem algebra_apbon2pownleqapownpbpowon2 (a b : ℝ) (n : ℕ) (h₀ : 0 < a ∧
0 < b) (h₁ : 0 < n) : ((a + b) / 2) ^ n ≤ (a ^ n + b ^ n) / 2 := by
  -- Step 1: Define the function f(x) = x^n
  let f : ℝ → ℝ := fun x => x ^ n
  -- Step 2: Show that f is convex
  have h2 : ConvexOn ℝ (Set.Ioi 0) f := by
    apply ConvexOn.pow
    · exact convexOn_id (convex_Ioi (0 : ℝ))
    · exact fun x hx => le_of_lt hx
  -- Step 3: Apply Jensen's inequality to f
  have h3 : f ((a + b)/2) ≤ (f a + f b)/2 := by
    have h4 : a ∈ Set.Ioi 0 ∧ b ∈ Set.Ioi 0 := by
      clear * - h₀
      obtain ⟨left, right⟩ := h₀
      apply And.intro
      · exact left
      · exact right
    have h5 : (a + b)/2 ∈ Set.Ioi 0 := by
      clear * - h₀
      obtain ⟨left, right⟩ := h₀
      (field_simp)
    have h6 : (1/2 : ℝ) + (1/2 : ℝ) = 1 := by
      clear * -
      (linarith)
    clear * - h2 h4 h5 h6
    rw [ConvexOn] at h2
    have h7 := h2.2 h4.1 h4.2 (by positivity) (by positivity) h6
    simp only [smul_eq_mul] at h7
    convert h7 using 1
    · ring_nf
    · ring
  -- Step 4: Rewrite using definition of f
  have h7 : ((a + b)/2) ^ n ≤ (a ^ n + b ^ n)/2 := by
    clear * - h3
    exact h3
  -- Final conclusion
  exact h7
```

## J.2 Solving a Putnam Problem on Real Analysis

DSP+ successfully solves a real analysis problem from the PutnamBench. We include it here as part of our results. This suggests that all three LLMs we employed demonstrate a genuine grasp of the subject beyond high school level. Furthermore, this highlights the general applicability of our framework.

**Statement**

```
open Metric Set EuclideanGeometry Filter Topology
theorem putnam_1970_b3 (S : Set (ℝ × ℝ)) (a b : ℝ) (hab : a < b) (hS : ∀ s
∈ S, s.1 ∈ Ioo a b) (hSclosed : IsClosed S) : IsClosed {y | ∃ x : ℝ, ⟨x,y⟩ ∈
S} := by
```

**Draft**

<think>...</think>
**Step 1:** Let $(y_n)$ be a sequence in $T$ converging to $y_0$.
**Step 2:** For each $n$, there exists $x_n \in (a, b)$ such that $(x_n, y_n) \in S$.
**Step 3:** The sequence $(x_n)$ is bounded since $x_n \in (a, b) \subseteq [a, b]$.
**Step 4:** By the Bolzano-Weierstrass theorem, there exists a subsequence $(x_{n_k}, y_{n_k})$ converging to some $(x_0, y_0)$.
**Step 5:** $x_0 \in [a, b]$ because $x_{n_k}$ is in $[a, b]$.
**Step 6:** Since $S$ is closed, $(x_0, y_0) \in S$.
**Step 7:** By hypothesis, $(x, y) \in S$ implies $x \in (a, b)$, so $x_0 \in (a, b)$.
**Step 8:** Thus, $(x_0, y_0) \in S$, so $y_0 \in T$.
**Step 9:** Therefore, $T$ contains all its limit points, hence $T$ is closed.

**Proof**

```
open Metric Set EuclideanGeometry Filter
Topology
theorem putnam_1970_b3 (S : Set (ℝ × ℝ)) (a
b : ℝ) (hab : a < b) (hS : ∀ s ∈ S, s.1 ∈
Ioo a b) (hSclosed : IsClosed S) : IsClosed
{y | ∃ x : ℝ, ⟨x,y⟩ ∈ S} := by
  -- Step 1: Let (yₙ) be a sequence in T
converging to y₀
  rw [isClosed_iff_clusterPt]
  intro y₀ h
  -- simp only [mem_setOf] at h
  have h1 : ∃ (u : ℕ → ℝ), (∀ n, ∃ x : ℝ,
(x, u n) ∈ S) ∧ Tendsto u atTop (𝒩 y₀) :=
by
    simp only
[clusterPt_principal_iff_frequently,
mem_setOf_eq] at h
    obtain ⟨u, hu⟩ : ∃ u : ℕ → ℝ, (∀ n : ℕ
, ∃ x : ℝ, (x, u n) ∈ S) ∧ Tendsto u atTop
(𝒩 y₀)
    · rcases exists_seq_forall_of_frequently
h with ⟨u, hu⟩
      exact ⟨u, hu.2, hu.1⟩
    · exact ⟨u, hu⟩
  -- Step 2: For each n, there exists xₙ ∈
(a,b) such that (xₙ, yₙ) ∈ S
  rcases h1 with ⟨u, hu, hu_tendsto⟩
  have h2 : ∀ n, ∃ x ∈ Ioo a b, (x, u n) ∈
S := by
    intro n
    rcases hu n with ⟨x, hx⟩
    exists x
    exact ⟨hS ⟨x, u n⟩ hx, hx⟩
  -- Step 3: The sequence (xₙ) is bounded
since xₙ ∈ (a,b) ⊆ [a,b]
  choose x hx using h2
  have h3 : ∀ n, x n ∈ Icc a b := by
    intro n
    exact Ioo_subset_Icc_self (hx n).1
```

```
  -- Step 4: By Bolzano-Weierstrass, there
exists a convergent subsequence
  have h4 : ∃ (φ : ℕ → ℕ) (x₀ : ℝ),
StrictMono φ ∧ Tendsto (x ∘ φ) atTop (𝒩
x₀) := by
    have h4 := isCompact_Icc.tendsto_subseq
h3
    rcases h4 with ⟨x₀, hx₀, φ, hφmono,
hφlim⟩
    exact ⟨φ, x₀, hφmono, hφlim⟩
  -- Step 5: x₀ ∈ [a,b] because x_{n_k} is
in [a,b]
  rcases h4 with ⟨φ, x₀, hφ, hxφ⟩
  have h5 : x₀ ∈ Icc a b := by
    have : ∀ n, x (φ n) ∈ Icc a b := by
      intro n
      exact h3 (φ n)
    clear * - hxφ this
    apply isClosed_Icc.mem_of_tendsto hxφ
    exact eventually_of_forall this
  -- Step 6: Since S is closed, (x₀,y₀) ∈ S
  have h6 : (x₀, y₀) ∈ S := by
    have : Tendsto (fun n ↦ (x (φ n), u (φ
n))) atTop (𝒩 (x₀, y₀)) := by
      rw [nhds_prod_eq, tendsto_prod_iff']
      constructor
      · exact hxφ
      · apply hu_tendsto.comp
        exact hφ.tendsto_atTop
    have : ∀ n, (x (φ n), u (φ n)) ∈ S :=
by
      intro n
      exact (hx (φ n)).2
    rename_i this_1
    apply hSclosed.mem_of_tendsto this_1
    simp [this]
  -- Step 7: By hypothesis, x₀ ∈ (a,b)
  have h7 : x₀ ∈ Ioo a b := by
    clear * - hS h6
    exact hS _ h6
  -- Step 8: Thus y₀ ∈ T
  exact ⟨x₀, h6⟩
```

## K Draft Comparison of Concise Steps and No Format

**"Concise Steps" on the left, "No Format" on the right**

**Step 1:**
$29 \mod 7 = 1$
**Step 2:**
$5 \mod 7 = 5$
**Step 3:**
$29^{13} \mod 7 = 1^{13} \mod 7$
**Step 4:**
$1^{13} = 1$
**Step 5:**
$5^6 \mod 7 = 1$
**Step 6:**
$13 = 6 \times 2 + 1$
**Step 7:**
$5^{13} \mod 7 = (5^6)^2 \times 5^1 \mod 7$
**Step 8:**
$(1)^2 \times 5 \mod 7 = 5 \mod 7$
**Step 9:**
$(29^{13} - 5^{13}) \mod 7 = (1 - 5) \mod 7$
**Step 10:**
$1 - 5 = -4$
**Step 11:**
$-4 \mod 7 = 3$

---

To prove that $(29^{13} - 5^{13}) \mod 7 = 3$, we use modular arithmetic and Fermat's Little Theorem.

1. **Simplify** $29 \mod 7$**:**
$$29 \div 7 = 4 \text{ remainder } 1 \implies 29 \equiv 1 \mod 7$$
Therefore:
$$29^{13} \mod 7 \equiv 1^{13} \mod 7 = 1 \mod 7$$

2. **Simplify** $5 \mod 7$**:**
$$5 < 7 \implies 5 \equiv 5 \mod 7$$
To compute $5^{13} \mod 7$, apply Fermat's Little Theorem:

Since 7 is prime and $5 ot\equiv 0 \mod 7$, $5^6 \equiv 1 \mod 7$

Express 13 as $6 \times 2 + 1$:
$$5^{13} = (5^6)^2 \times 5^1 \equiv (1)^2 \times 5 \equiv 5 \mod 7$$

3. **Compute the difference modulo 7:**
$$(29^{13} - 5^{13}) \mod 7 \equiv (1 - 5) \mod 7 \equiv -4 \mod 7$$
Convert $-4$ to a positive residue:
$$-4 + 7 = 3 \implies -4 \equiv 3 \mod 7$$

**Conclusion:**
$$\boxed{3}$$
Thus, $(29^{13} - 5^{13}) \mod 7 = 3$.

## L Draft Comparison of QwQ-32B and DeepSeek-R1

In our early toy experiments, we provide the same draft prompt to both QwQ-32B and DeepSeek-R1, and compare their final outputs after thinking session. To exemplify, the prompt is attached below:

**Draft Prompt**

formal_statement:
theorem aime_1983_p2 (x p : $\mathbb{R}$) (f : $\mathbb{R} \to \mathbb{R}$) ($h_0$ : $0 < p \wedge p < 15$) ($h_1$ : $p \le x \wedge x \le 15$) ($h_2$ : f x = abs (x − p) + abs (x − 15) + abs (x − p − 15)) : $15 \le$ f x := by

Please provide an extremely detailed mathematical calculation following your thinking. Each step can only contain **one** equation without any explanation.

Here is an example:
### Step 1:
\[ x + y + xy = 80 \]
. . .
### Step 5:
\[ x + y + xy + 1 = 81 \]

QwQ consistently produces responses that follow the instructions, as shown below:

**Output of QwQ-32B**

### Step 1:
$$ f(x) = |x - p| + |x - 15| + |x - (p + 15)| $$
### Step 2:
$$ \text{Given } p \leq x \leq 15 \implies |x - p| = x - p $$
### Step 3:
$$ \text{Given } x \leq 15 \implies |x - 15| = 15 - x $$
### Step 4:
$$ \text{Given } x \leq 15 \text{ and } p > 0 \implies x \leq 15 < p + 15 \implies |x - (p + 15)| = (p + 15) - x $$
### Step 5:
$$ f(x) = (x - p) + (15 - x) + (p + 15 - x) $$

### Step 6:
$$ f(x) = (x - p) + (15 - x) + (p + 15 - x) $$
### Step 7:
$$ f(x) = x - p + 15 - x + p + 15 - x $$
### Step 8:
$$ f(x) = (x - x) + (-p + p) + (15 + 15) - x $$
### Step 9:
$$ f(x) = 0 + 0 + 30 - x $$
### Step 10:
$$ f(x) = 30 - x $$
### Step 11:
$$ \text{Given } x \leq 15 \implies -x \geq -15 \implies 30 - x \geq 15 $$
### Step 12:
$$ f(x) \geq 15 $$

By contrast, the output of DeepSeek-R1 does not strictly adhere to the instructions and fails to follow the one-shot formatting. Additionally, DeepSeek-R1 sometimes includes its own Lean proof, which is often incorrect, as demonstrated below:

Step 1:
$$ f(x) = |x - p| + |x - 15| + |x - p - 15| $$
Step 2:
For $( p \leq x \leq 15 )$,
$$ |x - p| = x - p $$
Step 3:
$$ |x - 15| = 15 - x $$
Step 4:
$$ |x - p - 15| = p + 15 - x $$
Step 5:
Combine terms:
$$ f(x) = (x - p) + (15 - x) + (p + 15 - x) $$
Step 6:
Simplify:
$$ f(x) = x - p + 15 - x + p + 15 - x $$
Step 7:
$$ f(x) = -x + 30 $$
Step 8:
Since $( f(x) = 30 - x )$ is decreasing on $( [p, 15] )$, the minimum occurs at $( x = 15 )$:
$$ f(15) = 30 - 15 = 15 $$

Step 9:
Therefore, for $( p \leq x \leq 15 )$,
$$ f(x) \geq 15 $$
```lean4
theorem aime_1983_p2 (x p : ℝ) (f : ℝ → ℝ)
(h₀ : 0 < p ∧ p < 15) (h₁ : p ≤ x ∧ x ≤ 15)
(h₂ : f x = abs (x − p) + abs (x − 15) + abs (x
− p − 15)) : 15 ≤ f x := by
  have h₃ : p ≤ x := by linarith
  have h₄ : x ≤ 15 := by linarith
  have h₅ : 0 < p := by linarith
  have h₆ : p < 15 := by linarith
  simp_all only [abs_of_nonneg,
abs_of_nonpos, sub_nonneg, sub_nonpos, h₃,
h₄, h₅, h₆]
  -- For p ≤ x ≤ 15, simplify the absolute
values to linear expressions
  -- |x − p| = x − p, |x − 15| = 15 − x, |x − p −
15| = p + 15 − x
  -- Combine these to get f(x) = −x + 30
  -- Since f(x) = 30 − x is decreasing on [p,
15], the minimum occurs at x = 15
  -- f(15) = 30 − 15 = 15
  linarith
```

Therefore, we do not use DeepSeek-R1 as the sketch model at first, as this phase requires strong instruction-following capabilities.

## M    Proof Comparison of Different Solutions

Since Kimina-Prover-Preview uses the problem `imo_1962_p2`, to compare with BFS-Prover, we also present DSP+ proof for this problem. As can be seen, our generated proof well follows human proof conventions at the `have` statement level, resulting in higher readability. Only during the subgoal proving phase, the BFS-style and symbolic interaction with Lean slightly reduces the readability.

```
theorem imo_1962_p2 (x : ℝ) (h₀ : 0 ≤ 3 - x) (h₁ : 0 ≤ x + 1) (h₂ : 1 / 2 <
Real.sqrt (3 - x) - Real.sqrt (x + 1)) : -1 ≤ x ∧ x < 1 - Real.sqrt 31 / 8
:= by
  -- Step 1: Domain conditions
  have h₃ : x ≤ 3 := by
    clear * - h₀
    (linarith)
  have h₄ : -1 ≤ x := by
    clear * - h₁
    (linarith)
  -- Step 2: Original inequality
  have h₅ : Real.sqrt (x + 1) + (1 : ℝ) / 2 < Real.sqrt (3 - x) := by
    clear * - h₂
    (linarith)
  -- Step 3: Squaring both sides
  have h₆ : (Real.sqrt (3 - x))^2 > (Real.sqrt (x + 1) + (1 : ℝ) / 2)^2 := by
    clear * - h₅
    ring_nf
    ring_nf at h₅ ⊢
    nlinarith [Real.sqrt_nonneg (3 - x), Real.sqrt_nonneg (1 + x)]
  -- Step 4: Expand squared terms
  have h₇ : 3 - x > (x + 1) + Real.sqrt (x + 1) + (1 : ℝ) / 4 := by
    nlinarith [Real.sq_sqrt h₁, Real.sq_sqrt h₀]
  -- Step 5: Simplify inequality
  have h₈ : (7 : ℝ) / 4 - 2 * x > Real.sqrt (x + 1) := by
    clear * - h₇
    (linarith)
  -- Step 6: Square again
  have h₉ : ((7 : ℝ) / 4 - 2 * x)^2 > x + 1 := by
    clear * - h₈
    norm_num at h₈ ⊢
    rw [← sub_pos] at h₈
    contrapose h₈
    rw [sub_pos]
    rw [not_lt] at h₈ ⊢
    apply le_sqrt_of_sq_le
    exact h₈
  -- Step 7: Expand and rearrange
  have h₁₀ : 4 * x^2 - 8 * x + (33 : ℝ) / 16 > 0 := by
    clear * - h₉
    (linarith)
  -- Step 8: Multiply by 16 to clear denominators
  have h₁₁ : 64 * x^2 - 128 * x + 33 > 0 := by
    clear * - h₁₀
    (linarith)
  -- Step 9: Find roots of quadratic
  have h₁₂ : x < (8 - Real.sqrt 31) / 8 ∨ x > (8 + Real.sqrt 31) / 8 := by
    clear * - h₁₁
    refine' or_iff_not_imp_left.2 fun h => _
    contrapose! h
    contrapose! h₁₁
    nlinarith [h₁₁, h, Real.sq_sqrt (by norm_num : (0 : ℝ) ≤ 31)]
  -- Step 10: Eliminate impossible case due to domain
  have h₁₃ : x < (8 - Real.sqrt 31) / 8 := by
    contrapose! h₁₂
    refine'
      ⟨by linarith [Real.sqrt_nonneg (x + 1), Real.sqrt_nonneg (3 - x)], by
        linarith [Real.sqrt_nonneg (x + 1), Real.sqrt_nonneg (3 - x)]⟩
  -- Final conclusion
  have h₁₄ : x < 1 - Real.sqrt 31 / 8 := by
    clear * - h₁₃
    (linarith)
  exact ⟨h₄, h₁₄⟩
```

### Proof of IMO-1962-P2 found by BFS-Prover.

```
theorem imo_1962_p2 (x : ℝ) (h₀ : 0 ≤ 3 - x) (h₁ : 0 ≤ x + 1) (h₂ : 1 / 2 <
Real.sqrt (3 - x) - Real.sqrt (x + 1)) : -1 ≤ x ∧ x < 1 - Real.sqrt 31 / 8
:= by
    constructor
    linarith
    rw [← sub_pos]
    field_simp [Real.sqrt_lt] at h₂ ⊢
    apply lt_of_le_of_lt
    rw [mul_comm]
    rw [sub_eq_add_neg]
    apply lt_of_le_of_lt
    rw [← lt_sub_iff_add_lt]
    ring_nf
    rw [← lt_sub_iff_add_lt]
    linarith [Real.sq_sqrt (by linarith : 0 ≤ 1 + x)]
    rw [Real.sqrt_lt (by norm_num)]
    rw [Real.sqrt_lt] <;> nlinarith
    norm_num at this
```

### Proof of IMO-1962-P2 found by Kimina-Prover Preview.

```
theorem imo_1962_p2 (x : ℝ) (h₀ : 0 ≤ 3 - x) (h₁ : 0 ≤ x + 1) (h₂ : 1 / 2 <
Real.sqrt (3 - x) - Real.sqrt (x + 1)) : -1 ≤ x ∧ x < 1 - Real.sqrt 31 / 8
:= by
  have h3 : -1 ≤ x := by
    nlinarith
  have h4 : Real.sqrt (3 - x) - Real.sqrt (x + 1) > 1 / 2 := by linarith
  have h5 : Real.sqrt (x + 1) ≥ 0 := Real.sqrt_nonneg (x + 1)
  have h6 : (7 / 4 - 2 * x) > Real.sqrt (x + 1) := by
    nlinarith [Real.sq_sqrt (show (0 : ℝ) ≤ 3 - x by linarith),
Real.sq_sqrt (show (0 : ℝ) ≤ x + 1 by linarith),
    Real.sqrt_nonneg (3 - x), Real.sqrt_nonneg (x + 1)]
  have h7 : (7 / 4 - 2 * x) ^ 2 > (Real.sqrt (x + 1)) ^ 2 := by
    nlinarith [h6, Real.sqrt_nonneg (x + 1)]
  have h8 : (7 / 4 - 2 * x) ^ 2 > x + 1 := by
    have h10 : (Real.sqrt (x + 1)) ^ 2 = x + 1 := by
      rw [Real.sq_sqrt]
      linarith
    nlinarith [h7, h10]
  have h9 : 64 * x ^ 2 - 128 * x + 33 > 0 := by
    nlinarith [h8]
  have h10 : x < 1 - Real.sqrt 31 / 8 := by
    by_contra hx
    push_neg at hx
    have h12 : Real.sqrt 31 > 0 := by
      apply Real.sqrt_pos.mpr
      norm_num
    nlinarith [sq_nonneg (x - (1 - Real.sqrt 31 / 8)), sq_nonneg (Real.sqrt
31), Real.sq_sqrt (show (0 : ℝ) ≤ 31 by norm_num),
      h12]
  exact ⟨h3, h10⟩
```

### Proof of IMO-1962-P2 found by DeepSeek-Prover-V2.

```
theorem imo_1962_p2 (x : ℝ) (h₀ : 0 ≤ 3 - x) (h₁ : 0 ≤ x + 1) (h₂ : 1 / 2 <
Real.sqrt (3 - x) - Real.sqrt (x + 1)) : -1 ≤ x ∧ x < 1 - Real.sqrt 31 / 8
:= by
  have h₃ : -1 ≤ x := by
    linarith
  have h₄ : Real.sqrt (3 - x) - Real.sqrt (x + 1) > 1 / 2 := by
    linarith
  have h₅ : Real.sqrt (3 - x) > Real.sqrt (x + 1) + 1 / 2 := by
```

```
      linarith
  have h₆ : 3 - x > x + 1 + Real.sqrt (x + 1) + 1 / 4 := by
    have h₆₁ : Real.sqrt (3 - x) > Real.sqrt (x + 1) + 1 / 2 := h₅
    have h₆₂ : 0 ≤ Real.sqrt (x + 1) := Real.sqrt_nonneg (x + 1)
    have h₆₃ : 0 ≤ Real.sqrt (3 - x) := Real.sqrt_nonneg (3 - x)
    have h₆₄ : 0 ≤ Real.sqrt (x + 1) * (1 / 2) := by positivity
    have h₆₅ : 0 ≤ Real.sqrt (3 - x) := Real.sqrt_nonneg (3 - x)
    have h₆₆ : 0 ≤ Real.sqrt (x + 1) := Real.sqrt_nonneg (x + 1)
    have h₆₇ : (Real.sqrt (x + 1) + 1 / 2) ≥ 0 := by positivity
    have h₆₈ : Real.sqrt (3 - x) > Real.sqrt (x + 1) + 1 / 2 := h₅
    have h₆₉ : (Real.sqrt (x + 1) + 1 / 2) ^ 2 < (Real.sqrt (3 - x)) ^ 2 :=
by
      gcongr
      <;> nlinarith [Real.sq_sqrt (show 0 ≤ x + 1 by linarith),
Real.sq_sqrt (show 0 ≤ 3 - x by linarith)]
    have h₆₁₀ : (Real.sqrt (x + 1) + 1 / 2) ^ 2 = (x + 1) + Real.sqrt (x +
1) + 1 / 4 := by
      nlinarith [Real.sq_sqrt (show 0 ≤ x + 1 by linarith), Real.sq_sqrt
(show 0 ≤ 3 - x by linarith),
        sq_nonneg (Real.sqrt (x + 1) - 1 / 2)]
    have h₆₁₁ : (Real.sqrt (3 - x)) ^ 2 = 3 - x := by
      rw [Real.sq_sqrt] <;> linarith
    nlinarith [Real.sq_sqrt (show 0 ≤ x + 1 by linarith), Real.sq_sqrt
(show 0 ≤ 3 - x by linarith)]
  have h₇ : 7 - 8 * x > 4 * Real.sqrt (x + 1) := by
    have h₇₁ : 3 - x > x + 1 + Real.sqrt (x + 1) + 1 / 4 := h₆
    have h₇₂ : 0 ≤ Real.sqrt (x + 1) := Real.sqrt_nonneg (x + 1)
    nlinarith [Real.sqrt_nonneg (x + 1)]
  have h₈ : x < 7 / 8 := by
    by_contra h
    have h₈₁ : x ≥ 7 / 8 := by linarith
    have h₈₂ : 7 - 8 * x ≤ 0 := by
      linarith
    have h₈₃ : 4 * Real.sqrt (x + 1) ≥ 0 := by positivity
    linarith
  have h₉ : 0 ≤ Real.sqrt 31 := by positivity
  have h₁₀ : 64 * x ^ 2 - 128 * x + 33 > 0 := by
    have h₁₀₁ : 7 - 8 * x > 4 * Real.sqrt (x + 1) := h₇
    have h₁₀₂ : 0 ≤ Real.sqrt (x + 1) := Real.sqrt_nonneg (x + 1)
    have h₁₀₃ : 0 ≤ 7 - 8 * x := by linarith
    have h₁₀₄ : 0 ≤ Real.sqrt (x + 1) := Real.sqrt_nonneg (x + 1)
    have h₁₀₅ : 0 ≤ 7 - 8 * x := by linarith
    have h₁₀₆ : (7 - 8 * x) ^ 2 > (4 * Real.sqrt (x + 1)) ^ 2 := by
      gcongr
    have h₁₀₇ : (4 * Real.sqrt (x + 1)) ^ 2 = 16 * (x + 1) := by
      nlinarith [Real.sq_sqrt (show 0 ≤ x + 1 by linarith), sq_nonneg
(Real.sqrt (x + 1) - 2)]
    have h₁₀₈ : (7 - 8 * x) ^ 2 > 16 * (x + 1) := by
      linarith
    nlinarith [Real.sq_sqrt (show 0 ≤ x + 1 by linarith), sq_nonneg (x - 1)]
  have h₁₁ : x < 1 - Real.sqrt 31 / 8 := by
    by_contra h
    have h₁₁₁ : x ≥ 1 - Real.sqrt 31 / 8 := by
      linarith
    have h₁₁₂ : 64 * x ^ 2 - 128 * x + 33 ≤ 0 := by
      have h₁₁₃ : x ≥ 1 - Real.sqrt 31 / 8 := h₁₁₁
      have h₁₁₄ : x ≤ 1 + Real.sqrt 31 / 8 := by
        by_contra h₁₁₄
        have h₁₁₅ : x > 1 + Real.sqrt 31 / 8 := by linarith
        have h₁₁₆ : 64 * x ^ 2 - 128 * x + 33 > 0 := h₁₀
        have h₁₁₇ : x < 7 / 8 := h₈
        nlinarith [Real.sqrt_nonneg 31, Real.sq_sqrt (show 0 ≤ 31 by
norm_num)]
      nlinarith [Real.sqrt_nonneg 31, Real.sq_sqrt (show 0 ≤ 31 by
norm_num),
```

```
          sq_nonneg (x - (1 - Real.sqrt 31 / 8)), sq_nonneg (x - (1 +
  Real.sqrt 31 / 8))]
      linarith [h₁₀]
    exact ⟨h₃, h₁₁⟩
```

**Comparison with BFS-Prover.** Our observations are consistent with Kimina-Prover Preview: the proof generated by BFS is very difficult to understand, while both our code and Kimina-Prover Preview are relatively more aligned with human proof writing habits.

**Comparison with Kimina-Prover Preview.** A comparison between Kimina-Prover-Preview's open-source solutions and our solutions reveals a non-overlapping subset of solved problems. Specifically, Kimina-Prover-Preview's solutions include 4 problems that DSP+ does not solve, while our ensemble setting solves 11 problems unsolved by Kimina-Prover-Preview. This suggests a notable difference in problem-solving styles between the two approaches. Below, we present one example of a problem solved by Kimina-Prover Preview but not by our method.

```
theorem amc12a_2020_p4 (S : Finset ℕ) (h₀ : ∀ n : ℕ, n ∈ S ↔ 1000 ≤ n ∧ n
≤ 9999 ∧ (∀ d : ℕ, d ∈ Nat.digits 10 n → Even d) ∧ 5 | n) : S.card = 100 :=
 by
  have h1 : S = Finset.filter (fun n => 1000 ≤ n ∧ n ≤ 9999 ∧ (∀ d : ℕ, d ∈
Nat.digits 10 n → Even d) ∧ 5 | n) (Finset.Icc 0 9999) := by
    ext n
    simp [h₀]
    <;> tauto
  rw [h1]
  native_decide
```

**Comparison with DeepSeek-Prover-V2.** Our method fails to solve 16 problems that DeepSeek-Prover-V2 solves, but also succeeds on 3 problems that DeepSeek-Prover-V2 does not. One of the 3 problems is imo_2019_p1 in Appendix H.

