# OpenReview forum: "Reviving DSP for Advanced Theorem Proving in the Era of Reasoning Models"
_NeurIPS.cc/2025/Conference — NeurIPS 2025 poster_

### Official Review · Reviewer_hncJ · 2025-06-19

**Clarity:** 3
**Significance:** 2
**Originality:** 2
**Rating:** 4
**Confidence:** 3

**Summary:**

This paper revisits a technique called Draft, Sketch, Prove (DSP) on the performance of formal theorem proving with more advanced reasoning models. The authors identify that the original DSP framework has three independent phases that do not coordinate with each other. The authors propose DSP+ by tightly connecting the phases. Specifically, the draft phase leverages advanced reasoning models like DeepSeek-R1 while still making the response concise, the sketch phase checks for syntactically incorrect statements and repairs them and finally prove phase uses a more advanced BFS-prover. The results show that DSP+ attains strong performance close to SOTA DeepSeek Provers while using fewer inference tokens.

**Questions:**

1. Is it possible to integrate the techniques of DSP+ on top of the best models for proving such as DeepSeek-Prover-V2 671B and see if new SOTA results can be established?
2. For the ablation of components, how does "Draft + Sketch" prove theorems? Is it just with the common tactics?
3. In Figure 4, it seems that the No Format draft mode is better than Concise Steps, why not use this as the default and report main results with this setting?
4. Are authors fully committed to release all code to reproduce the main results? I believe a paper of such type needs to release code in order to benefit the community.
5. What is the iterative repair process discussed on Line 154?

**Ethical Concerns:**

["NO or VERY MINOR ethics concerns only"]

**Final Justification:**

I have read the authors’ rebuttal as well as the comments from Reviewer eJnA. Overall, I concur that the work is largely empirical and engineering-focused. Nevertheless, I recognize the substantial performance improvements achieved and appreciate the authors’ commitment to open-sourcing their work. I also hope that future methods will consider this paper as a baseline, rather than the original DSP. My perspective may be somewhat subjective, as I am very familiar with DSP.

**Limitations:**

The limitations section is very helpful and the authors provide various failure mode for their framework.

**Paper Formatting Concerns:**

N/A.

**Quality:**

3

**Strengths And Weaknesses:**

Strengths
- The paper revisits baselines that benefit from more advanced LLMs. This is crucial since baselines should be kept up to date in order to be fairly compared.
- The method does not require additional training, which is easy to adopt for practitioners.

Weaknesses
- Although the number of inference tokens is fewer than some baselines, the framework has a sequential workflow with different models which could make wall-clock time or resource requirement larger than some baselines.
- More weaknesses can be found in the Questions section.

---

> ### Author Rebuttal · Authors · 2025-07-31
>
> ### **Clarifying time and resource requirement comparison**
>
> While sequential workflows may incur higher latency per problem compared to one-shot generators, our framework remains highly competitive due to several key optimizations. First, by maintaining a persistent session for the ensemble tree search (`lean_copilot`, `aesop`), we avoid the repetitive overhead of launching new Lean processes, a feature not found in many prior works. Furthermore, we reuse the compiled header environment, saving approximately 10 seconds per verification call. Finally, our system supports high-throughput parallelism—up to 112 workers can run simultaneously.
>
> Regarding resource efficiency, our approach offers significant advantages. It is completely training-free, avoiding the significant GPU and time costs required for post-training or fine-tuning. It also consumes fewer inference tokens—a hardware-independent advantage that directly reflects reduced latency and GPU utilization. Moreover, our experiments—conducted mostly using readily available APIs—require only a 7B parameter proof model, making it more cost-effective and accessible than recent baselines of 72B or 671B that lack commercial APIs.
> ### **Integrating DSP+ with DeepSeek-Prover-V2 671B**
>
> Integrating state-of-the-art whole proof generators like DeepSeek-Prover-V2 671B into the proving phase of our framework is highly feasible. However, its immediate adoption is impractical for us due to current computational resource limitations and the lack of public APIs. We plan to explore this integration into our future system once these constraints are alleviated. Notably, DeepSeek-Prover-V2's technical report indicates that it already incorporates a DSP-like pipeline for data synthesis prior to reinforcement learning, suggesting an internalization of this reasoning process.
>
> ### **Clarifying how "Draft + Sketch" prove theorems**
>
> In this setup, the Sketch model (DeepSeek-V3 671B) is directly prompted to output the final, complete Lean code directly given the natural language draft. This means it produces tactic-level code without any "sorry" placeholders, unlike the standard sketch model.
>
> ### **Clarifying why not use No Format draft mode as the default (cf. Figure 4)**
>
> In our initial experiments, the performance of "No Format" and "Concise Steps" under smaller-scale tests (and up to pass@64 in Fig. 4) is nearly identical. Therefore, we chose "Concise Steps" as the default because it produced more human-readable proof drafts (see Appendix K).
>
> ### **Code release**
>
> Yes. We are committed to open-sourcing both the code and data once the review is finalized.
>
> ### **Clarifying the iterative repair process discussed on Line 154**
>
> The iterative repair process involves three key steps:
>
> 1. We parse the Lean code into a tree structure, using indentation to define its hierarchy.
> 2. The Lean compiler then identifies all lines containing errors.
> 3. For each error, we prune the corresponding node and its entire subtree from this code tree.
>
> This automated, non-destructive repair method offers greater robustness compared to simple truncation. We'll elaborate on this process further in the revision.

---

> > ### Comment · Reviewer_hncJ · 2025-08-05
> > **Response to Authors**
> >
> > Thank you to the authors for their rebuttal; I have no further questions and I will maintain my rating.
> >
> > I have read the authors’ rebuttal as well as the comments from Reviewer eJnA. Overall, I concur that the work is largely empirical and engineering-focused. Nevertheless, I recognize the substantial performance improvements achieved and appreciate the authors’ commitment to open-sourcing their work. I also hope that future methods will consider this paper as a baseline, rather than the original DSP. My perspective may be somewhat subjective, as I am very familiar with DSP.

---

### Official Review · Reviewer_GGi1 · 2025-06-30

**Clarity:** 4
**Significance:** 3
**Originality:** 3
**Rating:** 5
**Confidence:** 3

**Summary:**

This paper designs a novel formal theorem prover by advancing the Draft, Sketch and Prove framework. The model integrates neural-symbolic tools with LLMs. In contrast to prior works, LLMs are introduced in each phase, and integrated with neuro-symbolic tree search methods via built-in implementation in Lean4. The framework is free of training or finetuning, while achieves comparable performance to frontier theorem provers based on post-hoc training of LLMs. Ablation studies validate the robustness of design, and efficiency w.r.t. inference tokens.

**Questions:**

1. Since the most advanced formal prover generates whole proof, is it possible to integrate whole proof generators to the Prove phase, to replace or complement the BFS-prover?

**Ethical Concerns:**

["NO or VERY MINOR ethics concerns only"]

**Final Justification:**

Considering the feedback from the authors, I maintain my vote for acceptance. The paper has made the following contributions.

S1. The paper presents a refreshing view of the capability of the DSP framework with strong empirical results. It addresses the first effective integration of neural symbolic tools with LLMs.

S2. Thank the author for their response to W1. In summary, the proposed DSP framework is training-free in the sense that it utilizes and combines off-the-shelf models and produces a significant boost of performance against each component.

**Limitations:**

yes

**Quality:**

4

**Strengths And Weaknesses:**

### Strengths
1. As the most significant implication, the paper demonstrates that incorporating multiple existing LLMs and step provers into a delicately designed DSP framework can result in a significantly more powerful formal theorem prover. It increases BFS-Prover's accuracy on MiniF2F from 73% to 84%. Notably, the framework is free of post-hoc training of LLMs, while achieving comparable performance to the frontier large-scale whole-proof generator based on RL. The training-free feature is particularly valuable given the scarcity of formal proof samples.

2. The proposed method sharply improves over previous DSP frameworks (39%-84%). The improvement is mostly attributed to the prove stage, which features a novel built-in integration of LLM-based step prover with neuro-symbolic tactic tree searcher.

3. Sufficient ablation studies validate the synergy of all three component (DSP). The framework is flexible and modular, while being robust to the choice of LLMs for each phase.

### Weaknesses
1. The paper is not absolutely free of model training, since the step prover is post-trained from LLMs. However, the performance boost over  the step prover is significant enough.

---

> ### Author Rebuttal · Authors · 2025-07-31
>
> ### **Clarifying training-free of the framework**
>
> We apologize for the misunderstanding. When we refer to "training-free," we are highlighting that the DSP+ framework operates entirely with "off-the-shelf" models throughout its three stages. This approach contrasts with the conventional workflow of data collection followed by model fine-tuning. We will further clarify it in the next version.
>
> ### **Integrating SoTA whole proof generators to the Prove phase**
>
> It is quite possible to integrate SoTA whole proof generator (DeepSeek-Prover-V2 671B) into the prove phase of our framework. However, DeepSeek-Prover-V2 671B are currently impractical for us due to limited computational resources and the absence of public APIs. We intend to explore its integration into our future system once these constraints are resolved. Furthermore, its technical report indicates that DeepSeek-Prover-V2 already incorporates a DSP-like pipeline for data synthesis prior to reinforcement learning, suggesting that the model may have internalized this reasoning process.

---

> > ### Comment · Reviewer_GGi1 · 2025-08-09
> >
> > Considering the feedback from the authors, I maintain my vote for acceptance. The paper has made the following contributions.
> >
> > S1. The paper presents a refreshing view of the capability of the DSP framework with strong empirical results. It addresses the first effective integration of neural symbolic tools with LLMs.
> >
> > S2. Thank the author for their response to W1. In summary, the proposed DSP framework is training-free in the sense that it utilizes and combines off-the-shelf models and produces a significant boost of performance against each component.

---

> ### Comment · Area_Chair_kB33 · 2025-08-04
> **Friendly Reminder to Acknowledge or Update Your Review**
>
> Dear Reviewer GGi1,
>
> Thank you for your time and effort in reviewing the submissions and providing valuable feedback to the authors.
>
> If you haven't already done so, we kindly remind you to review the authors' rebuttals and acknowledge them by clicking the "Mandatory Acknowledgement" button at your earliest convenience. This step ensures efficient communication and helps finalize the process smoothly.
>
> We sincerely appreciate your dedication and collaboration.
>
>
> Best,
> Your AC

---

### Official Review · Reviewer_eJnA · 2025-07-03

**Clarity:** 1
**Significance:** 2
**Originality:** 1
**Rating:** 2
**Confidence:** 4

**Summary:**

The paper revisits the Draft-Sketch-Prove approach to formal theorem proving in the lean language. That is: given a lean statement we wish to prove we 1) prompt a model first to make a natural language proof draft, 2) prompt a model to extract formal goals conditional on this draft (which is roughly formatted as a partially completed lean proof with placeholder statements that need to be filled in) and 3) we use a llm + search algorithm to fill in the missing details of the previous step. This is all prior work - the current paper extends this with various improvements, most notably 1) using reasoning model for the draft step, 2) better prompting and post processing of the sketch and 3) integrating with a tree search method (aesop) rather than the simpler heuristic “sledgehammer” algorithm used in the original DSP paper.

**Questions:**

Q1) Can you please clarify the numbers on the rhs of figure 1? Why the + ? Why the 1024 vs 8192? Make the xB explicit as params in billions. Thanks!
Q2) Can you please list in a precise and unambiguous manner what the contributions are? My guess is that it is a) the use of chain of thought, b) using a newer and better base LLM, c) some heuristics to clean the sketches, d) prompting the model to make a concise sketch, e) masking generated lines of the sketch that produce lean errors, f) explicit specification in the subgoal hypothesis of the draft step.
Q2b) What does f) in Q2 mean?
Q3) Might # tokens be a better x axis for Fig 3?
Q4) Is the legend clear? I could not quite figure out exactly what was presented.

Minor points
Line 17 fewer budgets -> reword + state of the arts -> reword

**Ethical Concerns:**

["NO or VERY MINOR ethics concerns only"]

**Final Justification:**

I have considered everything already, thanks.

**Limitations:**

Yes

**Paper Formatting Concerns:**

Yes

**Quality:**

2

**Strengths And Weaknesses:**

Strengths

The overall problem is important. The experimental results are strong, only beaten out by the bespoke and super strong RL fine tuned DeepSeek Prover. The authors appear to be very talented at making the LLM pipeline really shine.

Weaknesses

The paper has relatively little technical novelty, but is rather a well executed application of lots of small known ideas. It presents a nice engineering effort, but this may not be appropriate for the NeurIPS main conference, and may be better suited to a more practical workshop track or so.

The presentation somewhat obscures the actual novelty in the paper. For example, the authors repeat the phrase “fine grained and integrated neuro symbolic enhancement” as a contribution, but this is rather vague. On line 64 the explanation for the good results is also rather vague.

The contributions such as certain prompt tweaks are somewhat tied to the specific LLM and not especially valuable scientifically.

---

> ### Author Rebuttal · Authors · 2025-07-31
>
> ### **Clarifying Figure 1**
>
> In RHS of Figure 1, each line represents "model size" x "inference tokens" x "inference times". For example, DeepSeek-Prover-V2 uses 671B model, and generate 8192 solutions for one problem, with each costs about 6.7K tokens on average.  The **"+"** in the rhs of Figure 1 indicates that our best configuration `(DeepSeek-R1-671B, DeepSeek-V3-0324-671B, BFS-Prover-7B)` uses the combination of two 671 billion-parameter models. And the two 671B models spend an average of 3.6k and 0.8k inference tokens in the miniF2F evaluation, respectively. Regarding the "1024 vs 8192" comparison, due to limited computational resources, we only report results for the 1024 budget. This budget still achieved superior performance compared to Kimina-Prover-Preview.
>
> ### **Clarifying technical contribution**
>
> The main concern of the reviewer is the technical contribution of our paper. We would like to clarify that, our contributions are twofold. First, we introduce the first method to implement a high-accuracy, DSP-based framework in Lean, integrating a series of systematic techniques (e.g., error line masking, hypothesis specification, and built-in tree search) within DSP. Our results significantly revitalize the effectiveness of DSP, which was previously noted as underperforming in Pantograph [1] and GoedelProver v1 [2]. Our framework reveals that even powerful natural language reasoning models, without task-specific fine-tuning, can achieve results comparable to state-of-the-art formal proof generation models when integrated with effective tree-search techniques. This finding aligns with recent advancements, such as DeltaProver [3], which secured a silver medal in the IMO 2025 competition.
>
> [1] Aniva, Leni, et al. "Pantograph: A machine-to-machine interaction interface for advanced theorem proving, high level reasoning, and data extraction in lean 4." *International Conference on Tools and Algorithms for the Construction and Analysis of Systems*. Cham: Springer Nature Switzerland, 2025.
>
> [2] Lin, Yong, et al. "Goedel-prover: A frontier model for open-source automated theorem proving." *arXiv preprint arXiv:2502.07640* (2025).
>
> [3] Zhou, Yichi, et al. "Solving Formal Math Problems by Decomposition and Iterative Reflection." *arXiv preprint arXiv:2507.15225* (2025).
>
> ### **Clarifying “prompt tied to specific LLM”**
>
> Our core contribution is the robustness of the DSP+ framework itself, not specific prompt text. To prove this, we demonstrate its general applicability across various LLMs. As shown in Figures 6 and 7, our framework functions effectively when using different models like DeepSeek-R1, GPT-4o, and QwQ-32B. The performance remains strong, confirming that the framework's design (including prompts) is robust and not over-tuned to a single model.
>
> ### **Explaining explicit specification**
>
> In the sketching stage, explicit specification significantly boosts the subsequent proving stage. Elaborately, this means that when an LLM converts a natural language draft into a formal proof sketch, it not only formalizes subgoals but also explicitly states any necessary hypotheses or assumptions for proving them.
>
> This is achieved by using the `prove_with [h2, h5]` notation (or similar), which clearly indicates the premises relevant to a given subgoal. For instance, consider the formal proof example in Appendix H (Page 22). If a subgoal were initially marked with `sorry` in the sketching stage, it would lead to an unconstrained and potentially excessive search for premises during the proving stage. However, by explicitly specifying `prove_with [h2, h3]` for that same subgoal, we effectively constrain the involved premises, thereby streamlining the proof search and making it far more efficient.
>
> ### **Using different x-axis in Figure 3**
>
> To benchmark the proving performance of various methods, we utilize Workflow Attempts Per Problem (equivalent to Pass@k). We also agree that the comparison of computational efficiency is necessary; therefore, we will include plots against total inference tokens in our revision.
>
> ### **Explaining legends in Figure 3**
>
> Figure 3 presents an ablation study demonstrating the necessity of each component in our framework. We use **DSP+ full (74.2%)** as a reference, which comprises QwQ-32B (Draft), DeepSeek-V3-0324 (Sketch), and BFS-Prover with Aesop (Prove). The comparison methods are detailed below:
>
> - **Draft + Sketch (22.5%)**: Skips the interactive Proving stage. The Sketch model is prompted to generate a complete proof directly.
> - **Sketch + Prove (71.7%):** Discards the natural language Draft stage; the Sketch model is directly prompted to generate a formal sketch for a given problem.
> - **QwQ-32B only (27.9%):** Uses only the Draft model to generate the final Lean code from the Lean problem statement.
> - **DeepSeek-V3-0324 only (26.6%):** Uses only the Sketch model to generate the final Lean code from the Lean problem statement.
> - **BFS-Prover with Aesop (49.6%):** Employs only the step-prover to solve the theorem given the Lean statement.
> - **Aesop only (35.2%):** A purely symbolic approach using Lean's built-in tactics, with no LLM involvement. The input is the Lean problem statement.
>
> ### **Minor points**
>
> Thanks for the comments. We will revise it in the next version.

---

> > ### Comment · Area_Chair_kB33 · 2025-08-04
> > **Friendly Reminder to Acknowledge or Update Your Review**
> >
> > Dear Reviewer eJnA,
> >
> > Thank you for your time and effort in reviewing the submissions and providing valuable feedback to the authors.
> >
> > If you haven't already done so, we kindly remind you to review the authors' rebuttals and acknowledge them by clicking the "Mandatory Acknowledgement" button at your earliest convenience. This step ensures efficient communication and helps finalize the process smoothly.
> >
> > We sincerely appreciate your dedication and collaboration.
> > Best,
> > Your AC

---

> > ### Comment · Reviewer_eJnA · 2025-08-04
> > **thanks**
> >
> > I appreciate the clarifications. My overall impression remains the same. I understand my view is a little subjective, and I appreciate that the other reviewers value this sort of contribution more highly than I do (i.e. which largley combines existing components). The quality of the engineering is good however and so I would be happy for the paper to be accepted, but I will leave my scores as they are.

---

> > > ### Author Response · Authors · 2025-08-06
> > > **Response to Reviewer on Novelty**
> > >
> > > Thank you for your thoughtful feedback and for acknowledging the value of our experimental results. We would like to respectfully clarify the novelty of our work, and explain why we believe it aligns well with NeurIPS's definition of originality.
> > >
> > > * On the novelty of our contributions:
> > >
> > > Even if one views our work as a composition of existing ideas, we believe the combination is far from trivial. To our knowledge, this is the first work demonstrating significant progress on theorem proving without any training, leveraging frontier LLMs for zero-shot reasoning. This insight aligns with recent strong works [1–3].
> > >
> > > Our proposed error line masking technique, when interpreted as a very straightforward form of code repair, it precedes several later works in Lean code repair [1–5]. Besides, our technique of hypothesis specification is a concrete technical innovation that advance upon the heuristics used in DeepSeek Prover v1.5 and v2.
> > >
> > > While we are not the first to replicate DSP in Lean, we are the first to uncover its true potential and significantly improve over prior accuracy baselines ([6,7]), which we believe provides meaningful direction for the field.
> > >
> > > More broadly, our findings open up a new and complementary path to theorem proving beyond large-scale training, which we hope will inspire further research.
> > >
> > > * On NeurIPS's definition of originality:
> > >
> > > We kindly refer to NeurIPS's guideline on originality (highlighted in https://neurips.cc/Conferences/2025/ReviewerGuidelines):
> > >
> > > "Originality does not necessarily require introducing an entirely new method. Rather, a work that provides novel insights by evaluating existing methods, or demonstrates improved efficiency, fairness, etc. is also equally valuable."
> > >
> > > In this light, we sincerely believe our work meets the bar for originality as defined by the conference, and we would be grateful if you could consider this in your assessment of the novelty and overall score.
> > >
> > > Thank you again for highlighting areas of improvement and for your valuable review.
> > >
> > > * References
> > >
> > > [1] Zhou, Yichi, et al. "Solving Formal Math Problems by Decomposition and Iterative Reflection." *arXiv preprint arXiv:2507.15225* (2025).
> > >
> > > [2] Baba, Kaito, et al. "Prover Agent: An Agent-based Framework for Formal Mathematical Proofs." *arXiv preprint arXiv:2506.19923* (2025).
> > >
> > > [3] Ospanov, Azim, Farzan Farnia, and Roozbeh Yousefzadeh. "APOLLO: Automated LLM and Lean Collaboration for Advanced Formal Reasoning." *arXiv preprint arXiv:2505.05758* (2025).
> > >
> > > [4] https://huggingface.co/blog/AI-MO/kimina-prover
> > >
> > > [5] https://blog.goedel-prover.com/
> > >
> > > [6] Lin, Yong, et al. "Goedel-prover: A frontier model for open-source automated theorem proving." *arXiv preprint arXiv:2502.07640* (2025).
> > >
> > > [7] Aniva, Leni, et al. "Pantograph: A machine-to-machine interaction interface for advanced theorem proving, high level reasoning, and data extraction in lean 4." *International Conference on Tools and Algorithms for the Construction and Analysis of Systems*. Cham: Springer Nature Switzerland, 2025.

---

> > > > ### Comment · Reviewer_eJnA · 2025-08-07
> > > > **thanks!**
> > > >
> > > > I remain happy to let the AC decide whether this is a "novel insight" worthy of NeurIPS.

---

### Decision · Program_Chairs · 2025-09-17

**Decision:**

Accept (poster)

**Comment:**

This paper develops a new version of Draft, Sketch, Prove (DSP, Jiang et al ICLR 2023), a method for formal theorem proving in interactive proof assistants. The new method, DSP+, achieves very strong performance, improving substantially upon the underlying models that it uses and exceeding the performance of methods based on different approaches to theorem proving (e.g., tactic-level tree search, full proof generation). This is in contrast to other attempts to adapt DSP to the Lean proof assistant, which showed only modest performance.

The three reviewers were split on their ratings, with two positive ratings (4, 5) and one negative rating (2). The negative review's primary concern was that DSP+ has limited novelty, amounting to a combination of many known components. However, the final judgement of novelty was left to the AC, and thus I will consider it closely here, since otherwise the reviewers were in favor of acceptance.

DSP+ has several interesting differences with DSP. DSP+ targets the Lean proof assistant rather the Isabelle proof assistant, which has several implications. Isabelle has access to powerful low-level automation (Sledgehammer) and typically follows a declarative proof style, which the original DSP took advantage of. Since Lean differs from Isabelle in these respects, it has been an open question whether a DSP-style approach is even suitable for Lean at all. Indeed, previous attempts have underperformed compared to alternatives, as I previously mentioned. Answering this question involved creating new components that were not present in the original DSP (e.g., hypothesis specification), figuring out many details, and justifying them through experiments. I view this as a valuable form of research.

The resulting system leads to a few interesting insights. For example, it is now clear that a DSP-style approach can achieve near state-of-the-art performance in Lean. This can impact the future direction of research in the area. Second, it shows that lower-level provers in the Lean toolchain (including tactic-style LLM provers such as BFS prover) are complementary to a higher-level sketching prover. This insight was not crystallized in the original DSP, since it relied on Sledgehammer in the Isabelle proof assistant.

Additionally, the authors use only open-weights models and will release the code, so DSP+ could serve as a new baseline or starting point for future research. Using only open-weights models and releasing code is becoming more rare in this sub-field, so this is a nice addition.

For these reasons, I agree with the evaluations of the positive reviewers. Overall, I believe that this paper offers an interesting and non-trivial extension of Draft, Sketch, Prove, and new insights that could impact the sub-fields of AI formal theorem proving and AI for mathematics. I recommend this paper for acceptance and think it would be a good addition to the NeurIPS program.